# Context-dependent perturbations in chromatin folding and the transcriptome by cohesin and related factors

Ryuichiro Nakato [1,6] ✉, Toyonori Sakata[2,3,4,6], Jiankang Wang [5], Luis Augusto Eijy Nagai[1], Yuya Nagaoka[1], Gina Miku Oba[1], Masashige Bando [2] & Katsuhiko Shirahige [2,3,4] ✉

Cohesin regulates gene expression through context-specific chromatin folding mechanisms such as enhancer–promoter looping and topologically associating domain (TAD) formation by cooperating with factors such as cohesin loaders and the insulation factor CTCF. We developed a computational workflow to explore how three-dimensional (3D) structure and gene expression are regulated collectively or individually by cohesin and related factors. The main component is CustardPy, by which multi-omics datasets are compared systematically. To validate our methodology, we generated 3D genome, transcriptome, and epigenome data before and after depletion of cohesin and related factors and compared the effects of depletion. We observed diverse effects on the 3D genome and transcriptome, and gene expression changes were correlated with the splitting of TADs caused by cohesin loss. We also observed variations in long-range interactions across TADs, which correlated with their epigenomic states. These computational tools and datasets will be valuable for 3D genome and epigenome studies.

The cohesin complex is crucial for gene transcription and chromatin folding in mammalian cells[1,2]. Cohesin colocalizes with the CCCTC-binding factor CTCF to function as an insulator[3]. In contrast, a small proportion of cohesin binds the genome independently of CTCF, regulating gene expression with tissue-specific transcription factors[4,5]. At least a subset of CTCF-independent cohesin mediates chromatin interactions between enhancer and promoter sites of active genes with mediator complexes[6]. Cohesin also participates in transcription elongation machinery by interacting with RNA polymerase II (Pol2)[7]. Mutations in the cohesin loader NIPBL and cohesin core subunits have been found in the human cohesinopathy Cornelia de Lange syndrome (CdLS), a multisystem developmental disorder[8], and in cancers[9,10]. Although the molecular mechanisms of CdLS are not fully understood,

the function of cohesin as a gene expression regulator is considered to be important[8,11].

Recent studies using whole-genome chromatin-conformation capture (Hi-C) have established the role of cohesin in the 3D organization of the genome[12,13]. Briefly, chromosomes are folded into megabase-scale TADs, the boundaries of which are strongly enriched for cohesin and CTCF[14]. TADs can be nested, and interactions between TADs (interTAD interactions) are rarer than those within TADs (intra-TAD interactions)[15,16]. The depletion of cohesin or the cohesin loader NIPBL causes a dramatic loss of TADs and chromatin loops throughout the genome[17,18], whereas CTCF depletion affects TAD boundaries and loops more locally[19]. These observations can be explained by the "loop extrusion" model, in which cohesin translocates along chromatin until

---

[1]Laboratory of Computational Genomics, Institute for Quantitative Biosciences, The University of Tokyo, 1-1-1 Yayoi, Bunkyo-Ku, Tokyo 113-0032, Japan. [2]Laboratory of Genome Structure and Function, Institute for Quantitative Biosciences, The University of Tokyo, 1-1-1 Yayoi, Bunkyo-Ku, Tokyo 113-0032, Japan. [3]Karolinska Institutet, Department of Biosciences and Nutrition, Biomedicum, Quarter A6, 171 77, Stockholm, Sweden. [4]Karolinska Institutet, Department of Cell and Molecular Biology, Biomedicum, Quarter A6, 171 77, Stockholm, Sweden. [5]School of Biomedical Sciences, Hunan University, Changsha, China. [6]These authors contributed equally: Ryuichiro Nakato, Toyonori Sakata. ✉e-mail: rnakato@iqb.u-tokyo.ac.jp; kshirahi@iqb.u-tokyo.ac.jp

it encounters CTCF, resulting in the formation of TADs[13,20]. This model also explains the depletion effects of cohesin unloading factors (WAPL and its binding partners PDS5A and PDS5B[2,21,22]), which prevent the release of cohesin from DNA and cause loop extension, resulting in the appearance of longer loops than usual[23,24]. The loop extension would be due to the "passing through" of CTCF sites by cohesin due to temporal dissociation of CTCF[23] or the physical clustering of neighboring CTCF boundaries called "traffic jam"[25]. Importantly, such extended loops are rare but also occur in wild-type cells[25]. This suggests that the dynamics of TAD and loop formation is not absolute, but rather stochastic, and depends on the amount of cohesin on chromatin as a result of its continuous loading and unloading (i.e., turnover)[26]. Cohesin turnover is also critical for the proper regulation of gene expression; its loss by the WAPL depletion resulted in the loss of cohesin localization and loops near several genes, leading to loss of gene expression[27]. A mutation in the cohesin deacetylase HDAC8 has also been found in CdLS, where increased cohesin acetylation results in low efficiency of cohesin turnover[28].

Despite extensive efforts, the detailed mechanism that results in hierarchical chromosome organization and the functional relationships that result in transcription regulation still need to be clarified[12]. Comprehensive loss of TADs/loops or loop extension has a limited impact on gene expression and does not cause the spread of histone modifications[17–19,23,29,30]. Moreover, cohesin and CTCF also localize within TADs without forming boundaries[5,31]. These results suggest a more complicated set of rules for chromatin structure formation and gene expression regulation by cohesin and related factors than the current models present. Although each cohesin-related factor has been studied using different cell lines by different laboratories, a study to explore how cohesin and its related factors collectively or individually regulate chromatin folding, gene expression, and the epigenome is needed.

To this end, we have generated 3D genome, transcriptome, and epigenome data before and after depletion of cohesin and related factors and conducted a large-scale comparative multi-omics analysis. To analyze our dataset, we developed a computational workflow to systematically compare multi-omics datasets, the main component of which is a package named CustardPy (Fig. 1a). CustardPy is primarily designed to compare multiple Hi-C samples to evaluate the variation of depletion effects across multiple proteins, and wraps several existing tools to cover the entire Hi-C process. Using our methodology, we comprehensively evaluated the similarities and differences in the effects of the depletion of individual factors, as summarized in Fig. 1b. Having confirmed the consistency of our results with those of previous studies that used different cell types and depletion methods (Figs. 2–4), we made the following observations. (1) Gene expression dysregulation that was correlated with splitting of TADs (i.e., TAD splits) was associated with the loss of cohesin (Fig. 5). (2) There was an imbalanced enrichment of cohesin binding on chromatin between active and inactive chromosome regions, which persisted even after CTCF depletion (Fig. 6). (3) CustardPy identified the context-specific pattern of inter-TAD interactions between depletions (Fig. 7). (4) Perturbation of long-range interactions was correlated with epigenomic states of loop anchors and TADs (Figs. 8 and 9). These computational tools and extensive datasets will be helpful for 3D genome and epigenome studies.

## Results

### Multi-omics data from a variety of cohesin-related depletions

Here we used human retinal pigment epithelium (RPE) cells to avoid the effects of aneuploidy or other genomic rearrangements. We used short interfering RNA (siRNA) to deplete cohesin (Rad21), cohesin loaders (NIPBL and Mau2), cohesin unloaders (WAPL, PDS5A and PDS5B), a boundary element (CTCF), and a cohesin acetyltransferase (ESCO1) individually and carried out two co-depletions (Rad21 and

NIPBL, PDS5A and B; Fig. 2a). We verified that most asynchronous cells were in G1 phase (Fig. S1a) and that the extent of depletion in most samples was sufficient for this analysis (Figs. 2b and S1b). Although the knockdown efficiency of PDS5B was not ideal, we included it as a reference in this study, because an incomplete loss of cohesin function causes the CdLS phenotype[8]. To minimize potential secondary effects, we evaluated the similarity and variability in depletion effects among factors rather than determining the function of each factor individually. We also generated a sample treated with the BET bromodomain inhibitor JQ1, as the bromodomain protein BRD4 is reported to interact with NIPBL and to be mutated in CdLS[32].

We used a 72-h treatment with siRNAs for most samples, but we also explored the effect of different treatment times (24, 48, and 120 h; Fig. S2a). Using these samples, we carried out in situ Hi-C, RNA-seq, and spike-in ChIP-seq data (Supplementary Data 1–3). We generated in situ Hi-C data with multiple independent replicates (31 samples, 14 billion paired-end reads in total). In the spike-in ChIP-seq analysis, we observed that 60–80% of the peaks in control cells were lost after siRNA (Fig. S2b).

We evaluated the overall similarity of our Hi-C samples and confirmed the sufficient similarity among replicates (Fig. 2c). We also found that the depletion effects could be categorized into four groups based on their siRNA targets (Fig. 2c): cohesin and loaders, CTCF, cohesin unloaders and acetyltransferase, and control and JQ1-treated samples. Depletion of cohesin unloaders and acetyltransferase showed a milder effect on chromosome structure than did depletion of cohesin loading and localization at CTCF sites. Having confirmed the sufficient similarity among replicates, we merged all replicates into a single deep Hi-C dataset for control, siRad21, siNIPBL (except for the 24-h treatment, at which the depletion was not sufficient; Fig. S2a), siCTCF, and siESCO1, resulting in up to 3 billion reads for each condition, for further analysis of these depletions.

### Comparative Hi-C analysis reveals diverse depletion effects on chromatin folding

We first evaluated the depletion effects on TADs and loops to verify that our results were consistent with those of previous studies that used different cell types and depletion methods. As expected, our Hi-C data showed a dramatic loss of TADs and loops after siRad21 or siNIPBL (Fig. 3a, b), consistent with the previous studies[17,18]. Co-depletion of Rad21 and NIPBL showed a more severe effect. Mau2 depletion led to a milder impact than did siNIPBL, possibly because some cohesin can be loaded without Mau2[23]. Although CTCF depletion strongly affected the number of loops, it had a limited effect on TAD numbers and intraTAD interactions as compared with cohesin depletion (Fig. 3a, b), confirming the function of CTCF as a boundary element[19,33].

Chromosomes are spatially segregated into active "compartment A" and inactive "compartment B" regions[34]. Such compartmentalization can be uncoupled from TAD formation; the compartmentalization is strengthened by the depletion of cohesin and loaders[17,18,23] but not by depletion of CTCF[19]. We observed a similar tendency in our data, as indicated by the "plaid pattern" (Fig. S3a) and quantitative compartment strength estimated by a saddle plot (Figs. 3b and S3b). This tendency was also indicated by the relative contact frequency of mapped reads (Fig. 3c). The depletion of cohesin and loaders diminished interactions at lengths consistent with TADs (-1 Mb), whereas the long-range interactions at a distance consistent with the compartment (-10 Mb) drastically increased. In contrast, depletion of CTCF, WAPL, PDS5A and B co-depletion or ESCO1 decreased long-range interactions, suggesting weakened compartmentalization. Depletion of PDS5A or PDS5B alone did not show a clear trend, suggesting the partly redundant function of PDS5A and PDS5B[24]. Finally, we did not observe notable compartment switching[35] in these samples (Fig. S3c).

We next explored the loop length distribution, which showed a distinct tendency from the relative contact probability (Figs. 3d and

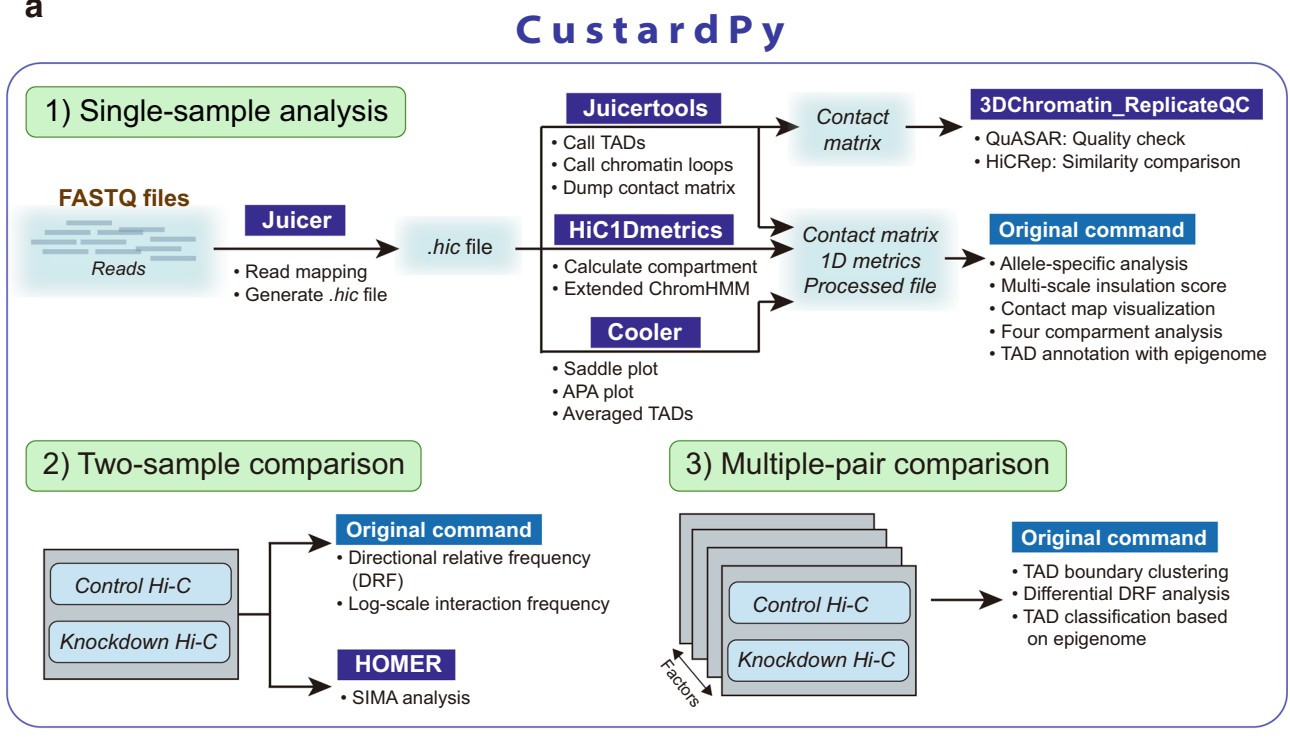

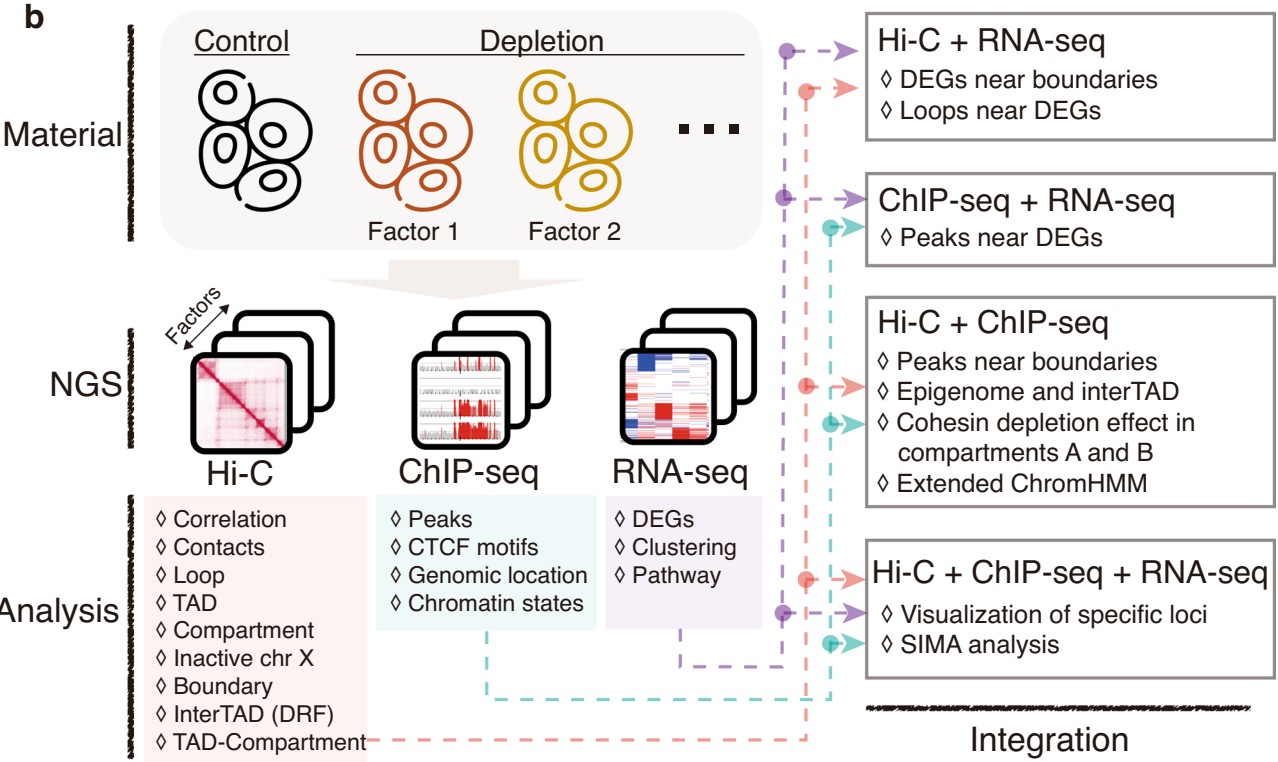

**Fig. 1 | Computational workflow of this study. a** Workflow of CustardPy. **b** Summary of the multi-omics analysis. DEG differentially expressed genes. SIMA a structured interaction matrix analysis.

S4a). After siRad21, most short loops were depleted, and the distribution peaked at a longer length (~1 Mbp) than the control (~400 kbp). siCTCF also resulted in depletion of short loops, which is less drastic compared with siRad21. siMau2 resulted in highly depleted long loops (~1 Mbp), and the distribution peaked at a shorter length than the control (~400 kbp). Based on the loop extrusion model, it is

likely that shorter loops were retained under the mild loss of cohesin after siMau2. After the depletion of unloaders (PDS5B, PDS5A and B, or WAPL), the peak distribution increased slightly relative to control samples (~500 kbp), consistent with their function as cohesin unloaders. siESCO1 also caused the appearance of longer loops, consistent with a previous report[36].

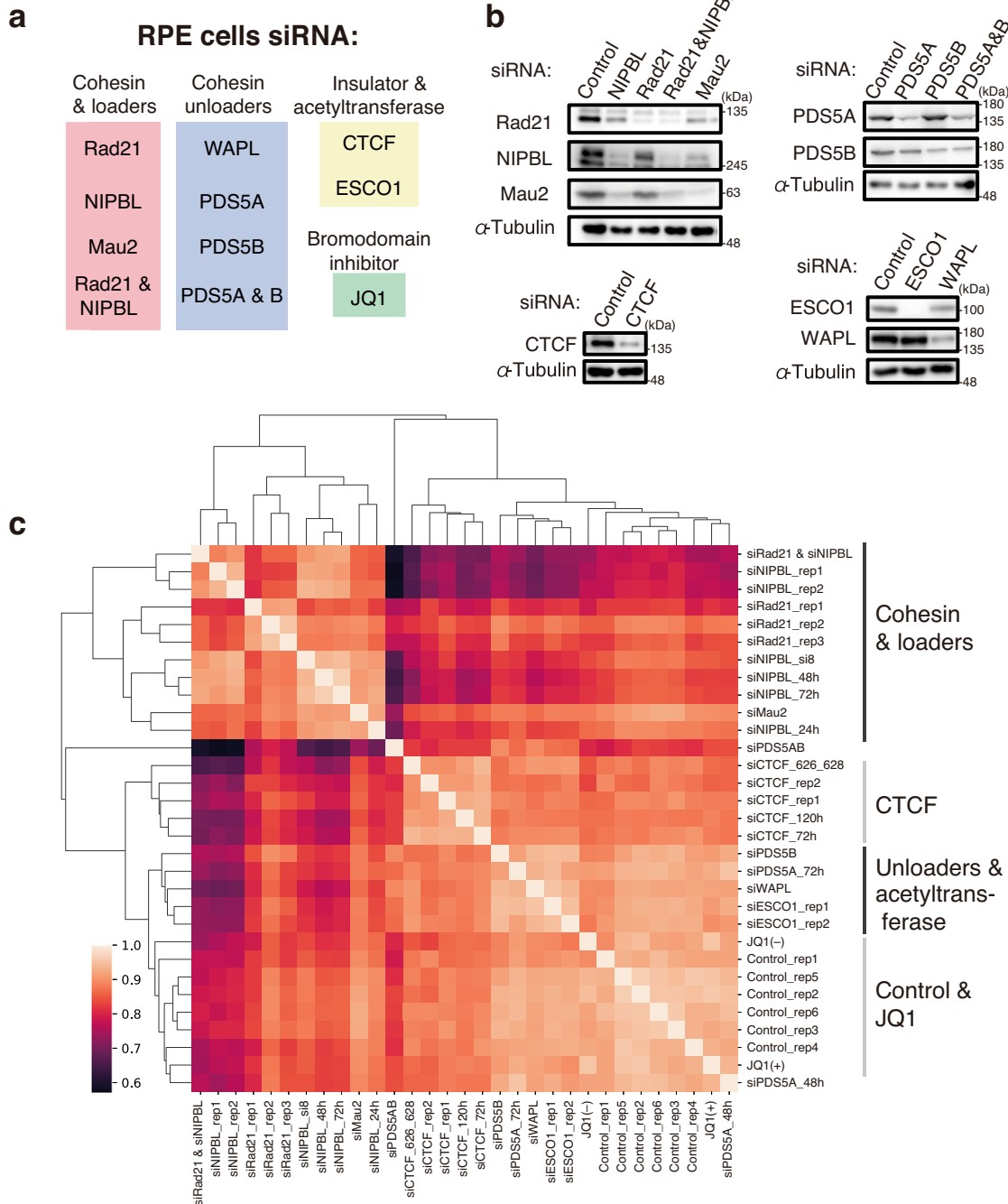

**Fig. 2 | Protein targets and their correlation to Hi-C data. a** Summary of protein targets. **b** Immunoblots after 72 h of siRNA. All blots were independently repeated for at least two biological replicates and similar results were obtained. **c** Correlation heatmap among all Hi-C samples based on stratum-adjusted correlation coefficients calculated by HiCRep[60]. Cells were treated with JQ1, designated as JQ1(+), to inhibit BET family proteins. Control for JQ1(+) are referred to as JQ1(−). Source data are provided as a Source Data file.

Additionally, we investigated the allele-specific depletion effect on chromosome X. Whereas the active chromosome X (Xa) forms the typical chromosome structure, inactive chromosome X (Xi) is partitioned into two megadomains, the boundary between which is affected by depletion of cohesin[37,38]. Our data did not show an explicit disruption of the megadomain boundary in Xi, possibly due to incomplete depletion by siRNA. However, we observed a difference in depletion effects between Xi and Xa, independent of depletion efficiency (Fig. S4). Xi showed a "coarser" plaid pattern than Xa, which was strengthened by siRad21 and siNIPBL. siCTCF showed an asymmetric tendency of interaction frequency between the megadomain

boundary and the two megadomains (black arrows), whereas Xa had no similarly clear chromosome-wide pattern.

In summary, our Hi-C analysis showed consistent tendencies with previous studies, confirming its reliability, and provided multiple new findings of diverse depletion effects on chromatin folding.

## Gene expression changes are correlated with direct cohesin binding

Next, we explored the depletion effect on gene expression and ChIP-seq peaks. We detected 2000–7000 differentially expressed genes (DEGs) for each sample (false discovery rate [FDR] <0.01; Figs. 4a and S5a). We

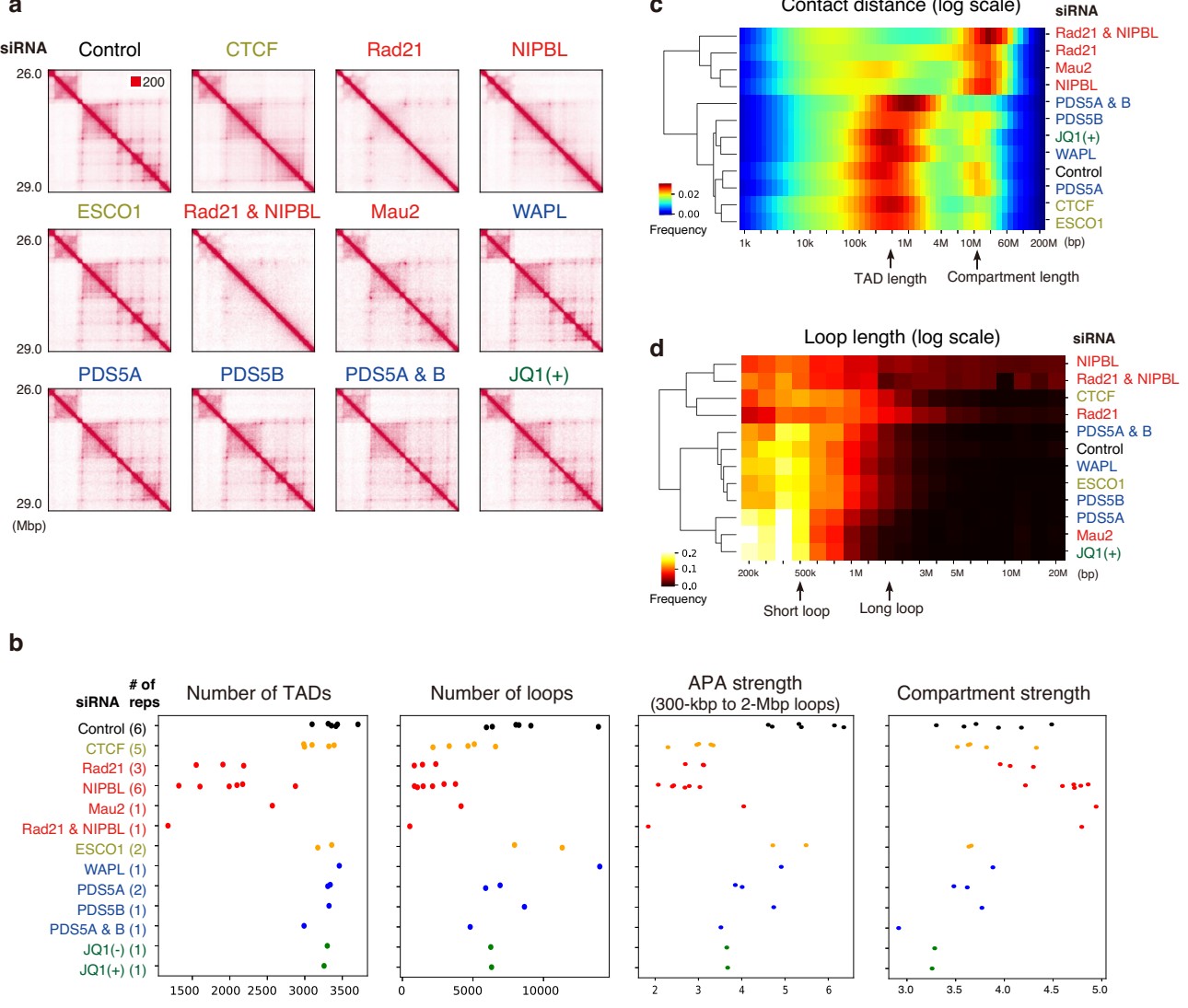

**Fig. 3 | Comparative Hi-C analysis to explore the variability of depletion effects.** **a** Normalized Hi-C matrix of a representative chromosomal region (chromosome 21, 26.0–29.0 Mb). **b** The number of TADs, loops, aggregate peak analysis (APA) strength (observed/expected ratio of the center) and compartment strength (from saddle plots; see also Fig. S3b). The dots indicate the Hi-C samples (including different treatment times). **c** Relative contact probability for the likelihood of a contact at increasing length scales (averaged across replicates). **d** Loop length distribution. The color indicates the fraction of loops compared to all loops. Source data are provided as a Source Data file.

selected the top-ranked 1,000 DEGs from all samples and merged them into a single DEG list (4240 genes in total). Pairwise comparisons showed an overlap of DEGs between cohesin and loaders and between individual unloaders (Fig. 4a, red boxes). siCTCF and JQ1 treatment showed less correlation with the others, suggesting their distinct roles in gene expression regulation.

To identify the pattern of expression dysregulation, we applied k-means clustering (k = 20) based on the overlap of up- and down-regulated genes among depletions (Fig. 4b and Supplementary Data 4). For example, clusters 6 and 10 represent down- and upregulated genes after cohesin and loader depletion, respectively. Gene ontology (GO) analysis suggested that cluster 6 was mainly enriched in "growth factor activity," consistent with slower growth under such depletions[39]. Clusters 9 and 18 contained down- and upregulated genes after NIPBL and unloader depletions, respectively. These DEGs were not observed after siRad21, suggesting DEGs from dysregulation of cohesin turnover. Their GO terms were correlated with fundamental functions related to the cytoskeleton and extracellular matrix. These diverse expression patterns suggest

that cohesin-related factors have distinct roles in gene expression regulation.

We then implemented a permutation test (n = 1000) to examine the overlap of the DEG loci with the ChIP-seq peaks and chromatin loops (Fig. 4c). Transcription start sites (TSSs) associated with the DEGs were enriched for Rad21 and Mau2 peaks in most clusters, suggesting that the DEGs were less likely to have been derived from a secondary effect. The exceptions were the downregulated genes after siWAPL (clusters 3 and 9), which were affected independently of cohesin binding, implying the indirect or unrelated regulation relative to cohesin. We also applied a TAD-boundary proximity analysis[29] and found that the siRad21 DEGs were less likely to be located around disrupted TADs than non-differential boundaries (Fig. S5b). These results suggest that expression dysregulation of these clusters was caused by the loss of Rad21 and Mau2 binding to each gene, rather than by region-wide effects caused by TAD disruption.

A certain amount of cohesin on the genome is acetylated by ESCO1, which protects cohesin from release by WAPL, resulting in more stable binding of cohesin to the genome[40]. Therefore, the

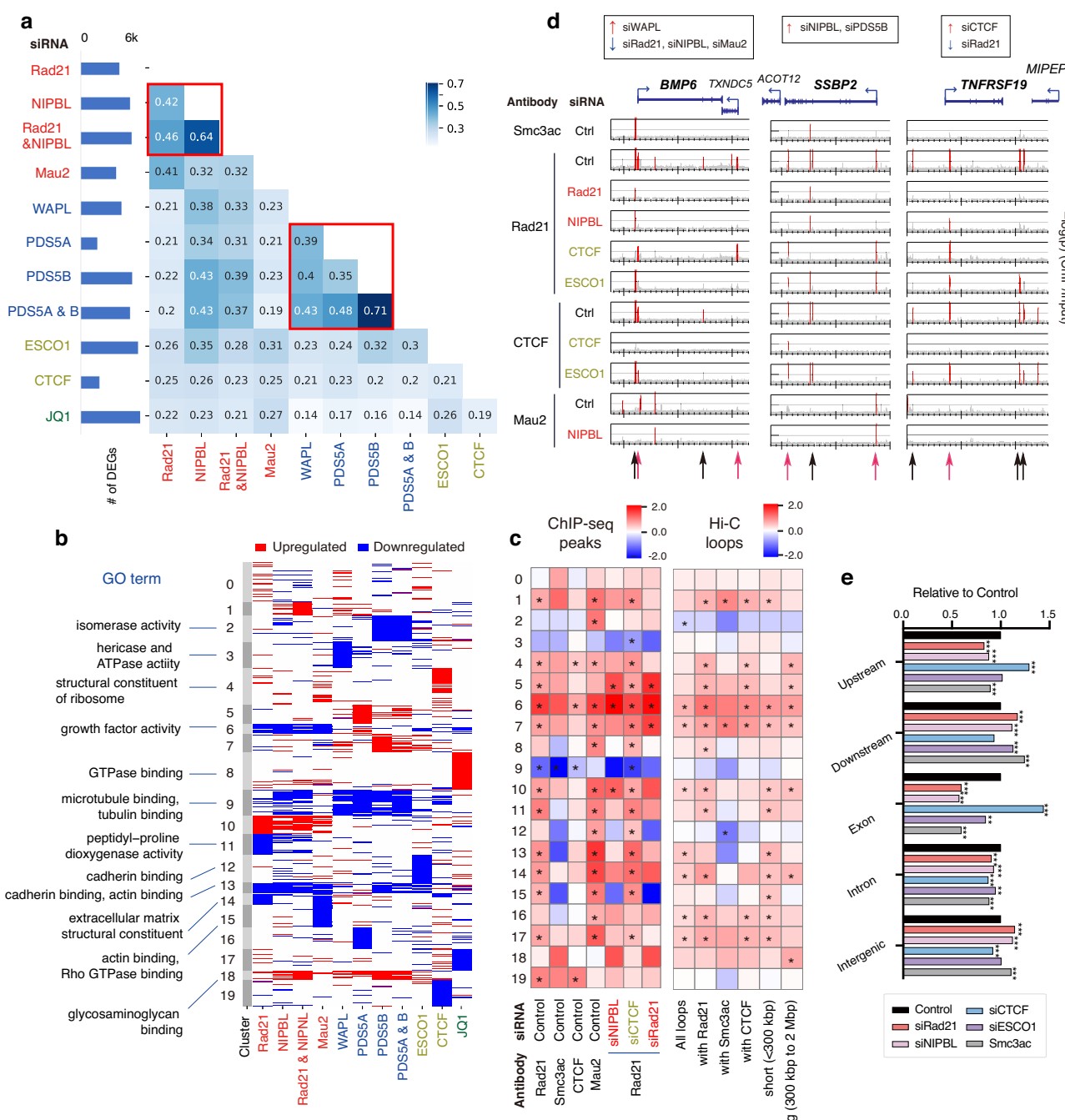

**Fig. 4 | Comparative analysis of siRNA effects on the transcriptome and epigenome. a** Left: Number of DEGs for each siRNA treatment (FDR < 0.01). Right: Correlation heatmap based on the Simpson index (showing overlap among top-ranked 1000 DEGs) for all sample pairs. **b** Clustering of all top-ranked DEGs (rows) and samples (columns) based on the overlap of up- and downregulated genes. The significant GO terms are also shown (left). See Fig. S6 for the full list of GO terms. **c** Log-scale relative enrichment of ChIP-seq peaks (left) and Hi-C loops (right) at the TSSs of DEGs in the clusters corresponding to (**b**) against all DEGs. * indicates the statistical significance of the enrichment or depletion

($p < 0.01$, two-tailed permutation test, $n = 1000$). **d** ChIP-seq distribution ($-\log_{10}(p)$, two-tailed binomial test, 100-bp bin) among top-ranked DEGs (*BMP6*, *SSBP2*, and *TNFRSF19*). The significantly enriched regions ($p < 10^{-4}$) are highlighted in red. Red and black arrows (bottom) indicate CTCF-independent and -dependent Rad21 peaks, respectively. **e** The relative enrichment of Rad21 peaks after depletions and of Smc3ac around genes, as compared with Rad21 (control). **p < 0.001; ***p < 0.0001, two-tailed Fisher's exact test against control. The exact p-values are shown in Source Data file. Source data are provided as a Source Data file.

acetylated cohesin sites (Smc3ac) would be more persistent even under siNIPBL and siRad21 than non-acetylated sites. This tendency can be observed in Rad21 peaks in siRad21, i.e., since stronger Rad21 peaks remain more under siRad21, the clusters where Rad21 peaks are enriched under siRad21 indicate containing more stronger Rad21 peaks at TSSs. The clusters enriched for Smc3ac and Rad21

under siRad21 are highly correlated (0, 5, 7, 10, 14, and 18). Loops mediated by Smc3ac were enriched in upregulated DEGs associated with siNIPBL and siPDS5B (clusters 1, 7, 18), whereas clusters not enriched for Smc3ac (clusters 12, 13, 15) were downregulated (Fig. 4c). This result also suggests the necessity of cohesin at TSSs for gene expression.

Figure 4d shows the ChIP-seq distribution around several top-ranked DEG loci, each having cohesin peaks around their TSSs. Rad21 peaks around TSSs were lost after siNIPBL, whereas they remained after siCTCF (red arrows), suggesting that cohesin binds at TSSs in a more CTCF-independent manner. In contrast, Rad21 peaks in BMP6's exon, SSBP2's intron, and the intergenic region were lost after siCTCF (black arrows). We confirmed that this tendency occurred genome-wide (Fig. 4e). siNIPBL significantly depleted cohesin peaks at upstream and exon regions, whereas siCTCF affected intron and intergenic regions. This result is reminiscent of a report using CTCF knockout mouse embryonic fibroblast (MEF) cells[41], in which cohesin accumulated near TSSs of active genes, where the cohesin loader is also located. In the absence of CTCF, cohesin would be more correlated with gene activity and cohesin loading sites. Our results suggest that cohesin positively regulates gene expression via direct binding at TSSs. In contrast, other mechanisms (e.g., turnover and acetylation) add to the diversity of this pattern of dysregulation.

## Quantitative classification of insulation levels among boundaries

To further study the depletion effects on chromatin folding, we compared a multi-scale insulation score[42] among samples (Fig. 5a–c). We found various patterns of insulation perturbation at TAD boundaries: (i) boundaries depleted by siRad21 and siNIPBL but not by siCTCF (cohesin-dependent); (ii) boundaries strengthened by siNIPBL and siRad21 (cohesin-separated); (iii) boundaries depleted by siRad21, siNIPBL, and siCTCF (all-dependent); and (iv) boundaries that were barely affected by any siRNA (robust). To quantify the observed patterns across the genome, we classified all boundaries into six categories based on their insulation scores (Fig. 5d and Supplementary Data 5). Whereas over half of the boundaries were annotated as "robust," indicating their stability against reduced amounts of a targeted protein, we also identified those subcategories of boundaries that were lost or gained after depletions. Depletion of unloader proteins did not show an explicit perturbation, which is consistent with their minimal influence on the number of TADs (Fig. 3b).

We then investigated the overlap among boundaries, ChIP-seq peaks, and DEGs (Figs. 5e and S7a). Whereas cohesin- and CTCF-dependent boundaries (i.e., boundaries that were lost after depletion) were enriched for loops and CTCF peaks, there were few DEGs, even though Pol2 peaks were enriched at cohesin-dependent ones. In contrast, cohesin-separated boundaries significantly overlapped with upregulated DEGs after the depletion of cohesin and loaders. At the boundaries, Mau2 peaks were strikingly enriched, but Rad21, CTCF, and loops were not, suggesting that these are cohesin loading points. This result suggests that the loss of cohesin loading at these points leads to an enhancement of insulation, which could potentially lead to dysregulation of gene expression in the surrounding region. As CTCF-separated boundaries also overlapped with DEGs (although not significantly), a gain of boundaries would be more highly correlated with upregulated DEGs than a loss of boundaries. These observations highlight the necessity of considering the boundary type based on the depletion effects of factors when investigating the association between chromatin folding and gene expression as regulated by cohesin.

In addition, CTCF-dependent and cohesin-separated boundaries occurred more frequently between compartments A and B, whereas cohesin-dependent ones occurred less frequently (Fig. S7b). This result implies that cohesin has a role in connecting neighboring TADs[18], especially those from compartments A and B. In contrast, CTCF is involved in partitions within compartment A and, to a lesser extent, compartment B.

## Cohesin is broadly distributed in the active compartment

We next explored the genomic regions that changed significantly after depletions across the genome to investigate the correlation between

the depletion effect on the epigenome and chromatin folding (Figs. 6a and S8a). To directly identify significant changes after depletion at the absolute level, we evaluated $-\log_{10}(p)$ values comparing ChIP reads of control and siRNA samples under spike-in normalization[43]. As reported in previous studies[17,19], broad histone marks (H3K36me3, H3K27me3, and H3K9me3) were not substantially affected, suggesting that the loss of TAD boundaries does not cause these histone modifications to spread. In contrast, H3K27ac marks were largely perturbed after NIPBL and Rad21 depletion, indicating that these marks can be affected by changes in intraTAD or interTAD interactions.

We next investigated the binding of cohesin and several related factors. Strikingly, we observed a broad decrease in cohesin binding after siRad21 and siNIPBL (Fig. 6a, purple arrows). In contrast, siCTCF depleted cohesin binding only at CTCF binding sites (Fig. 6a, red regions of CTCF in siCTCF, green arrows). This suggests that cohesin is located not only in peak regions as detected by peak calling but also in background regions more broadly, as assumed in the loop extrusion model[20].

We further investigated this tendency across the genome by dividing compartments A and B into "strong" and "weak" ones based on their compartment PC1 values (Fig. 6b). The significant depletion of cohesin in the background as noted above was not detected within compartment B regions (Fig. 6a, blue bars), suggesting that there are different amounts of cohesin between compartments A and B. Genome-wide, we observed a more significant amount of cohesin, particularly in "strong A" regions (Fig. 6b). This observation is also supported by the milder loss in Strong B and a more severe loss in Strong A in intraTAD interactions under siRad21 compared to siCTCF (Figs. 6c and S8b).

We also evaluated cohesin enrichment from an epigenomic perspective using "extended ChromHMM"[44] and found that cohesin accumulated to the highest levels at highly active sites (states 4 and 7, enriched for H3K27ac and H3K4me3; Fig. 6d). Heterochromatin regions enriched for H3K9me3 in compartment B showed subtle cohesin enrichment (state 14). In contrast, siCTCF did not show such a context-specific tendency (Fig. 6b, d), which indicates that the imbalance in the amount of cohesin between compartments A and B was retained even after the loss of CTCF-dependent boundaries. The imbalance was also observed in our biological replicate data (Fig. S8c) and also partly reported in a previous study that used calibrated ChIP-seq analysis of *Nipbl*-deleted mouse liver[18]. Consequently, our data showed that cohesin also accumulated in non-peak regions, more so in compartment A, in conjunction with gene activity. CTCF acts as an obstacle for cohesin translocation (resulting in sharp cohesin peaks at CTCF sites) but does not control the amount of cohesin on chromatin.

## InterTAD interactions are affected by context-specific depletion

In addition to the six boundary types (Fig. 5d), we also found long-range insulation boundaries that appeared after siNIPBL and siRad21 (-500 kbp; Fig. 5c, red rectangle). By visualizing the relative interaction frequency (Figs. 7a and S9a), we found that these insulation boundaries likely reflect the substantial depletion of interactions between an "active TAD" (Fig. 7a, A4, enriched by active markers H3K4me3 and H3K27ac, and Pol2) and an "inactive TAD" (Fig. 7a, B2, in compartment B, black arrows). Although a decreased interaction between active and inactive regions is consistent with compartmentalization strengthening[18], this depletion effect was more region specific and was not symmetrical (e.g., there was a milder effect between B2 and A5; Fig. 7a). Moreover, we also observed a difference even between siRad21 and siNIPBL on interTAD interactions (e.g., A3–B2; Fig. 7a, black rectangles), despite their very similar effects on TAD and loop structures. We were, therefore, interested in the variation in perturbations of interTAD interactions among different siRNA targets.

To identify the strong effect of depletions on interTAD interactions, we recently developed the directional relative frequency (DRF)

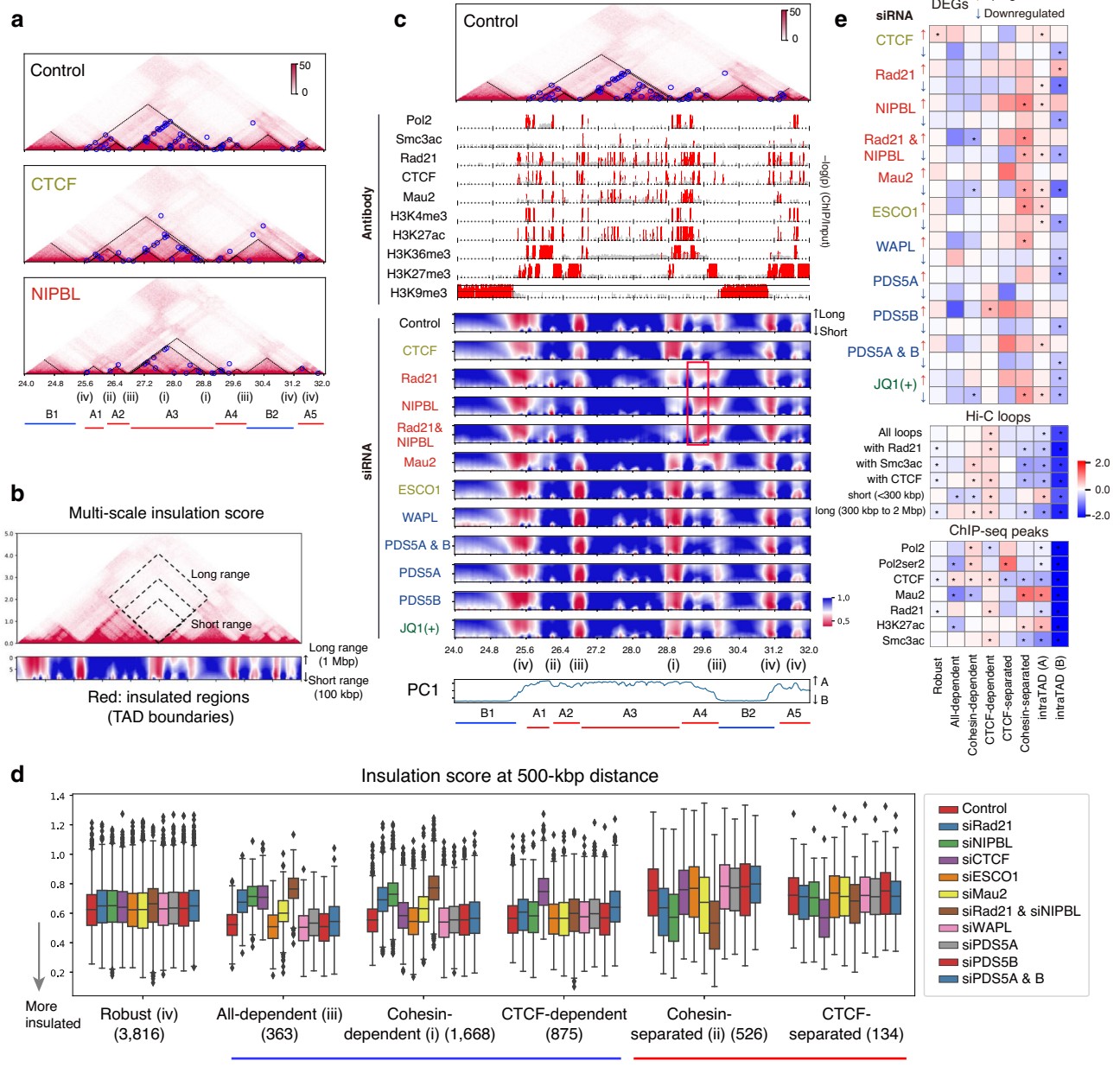

**Fig. 5 | Multi-scale insulation scores reveal the diversity of insulation perturbations at TAD boundaries. a, c** A representative chromosomal region (chromosome 21, 24.0–32.0 Mb). The numbers along the bottom (i–iv) indicate four of the six boundary types (see text). For clarity, the genomic region was manually classified into A1–5 (compartment A) and B1 and B2 (compartment B) based on the TAD structure. **a** Normalized Hi-C matrices. Black dashed lines and blue circles indicate TADs and loops, respectively. **b** Schematic illustration of multi-scale insulation score[42]. **c** Top: Hi-C heatmap of Control. Upper middle: ChIP-seq distribution (−log10($p$), two-tailed binomial test, 5-kbp bin). Red regions: $p < 10^{-3}$. Lower middle: Multi-scale insulation scores. Red regions indicate insulated regions (boundaries). The red rectangle indicates the long-range insulation boundaries present after siNIPBL. Bottom: Compartment PC1. **d** Insulation score distribution for six boundary types (whisker plots with median and interquartile range [IQR]). The number in parentheses below each boundary type indicates the number of boundaries identified. **e** Relative enrichment of DEGs (top), loops (middle), and ChIP-seq peaks (bottom) that overlapped the six boundary types and intraTAD regions for compartments A and B against all boundaries. *$p < 0.01$ (two-tailed permutation test, $n = 1000$). Source data are provided as a Source Data file.

approach[44], which evaluates the directional bias of long-range depletion effects (Fig. 7b). We scanned the whole genome and identified 241 regions in which DRF values changed significantly after cohesin or loader depletion (black ovals in Fig. 7c and Supplementary Data 6). Some of these changes consisted of a decrease across broad regions (C1 and C2), whereas other regions showed a decreased interaction on one side of a TAD (C3 and C4), reminiscent of the "stripe" structure, where a loop anchor site highly interacts with entire region of a TAD[45]. Whereas stripes were reported to be near super-enhancer regions[45],

the differential DRF regions in our data were often located at the points of transition for compartmental PC1 values (dashed lines in Fig. 7c, d). Because PC1 values around these transition points were not perturbed by siNIPBL (Fig. 7d), the substantial depletion in interTAD interactions is likely to be distinct from the strengthened compartmentalization. Moreover, these interactions often increased after siRNA of the cohesin unloader WAPL; therefore, the effect was inversely correlated with depletion of cohesin and loaders. We examined this tendency across all 241 regions using cosine similarity of the

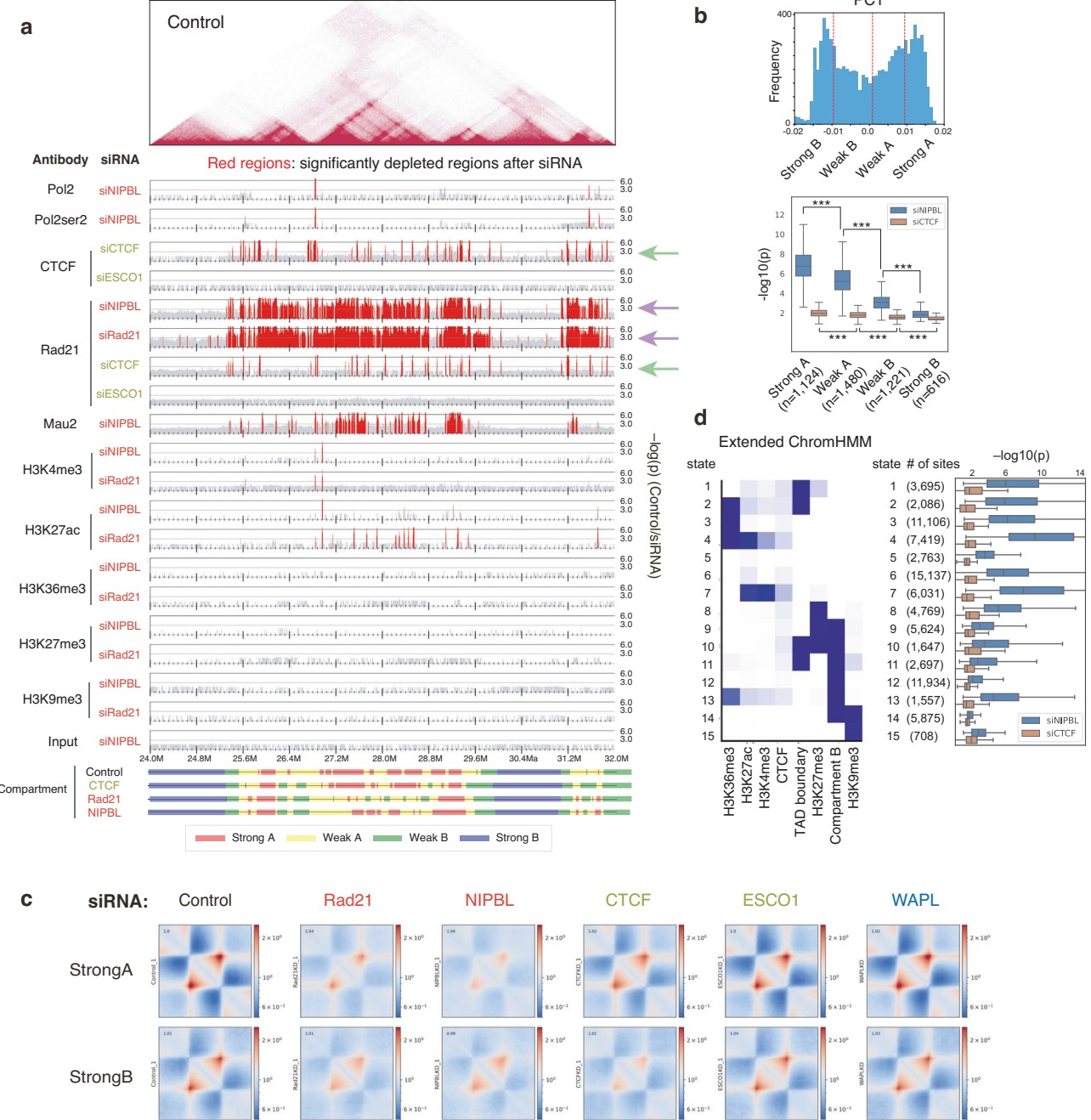

**Fig. 6 | Cohesin was broadly enriched in compartment A but was not enriched in compartment B. a** The depletion effect distribution from ChIP-seq data ($-\log_{10}(p)$, two-tailed binomial test, 5-kbp bin) on chromosome 21, 24−32 Mbp. The colored bars (bottom) indicate the four compartment types as determined by compartment PC1 values. **b** Top: PC1 value distribution of a control Hi-C sample and the definition of four compartment types. Bottom: $-\log_{10}(p)$ distribution (two-tailed binomial test, 5-kbp bin) for the depletion effect of siNIPBL and siCTCF on Rad21 ChIP-seq in the four compartment types (box plots with median and IQR).

***$p < 0.0001$ (two-tailed Mann−Whitney U-test). The exact p-values are shown in Source Data file. **c** Average interactions (observed/expected) in Strong A and Strong B TADs for representative samples. **d** Left: Fifteen chromatin states using extended ChromHMM, which include compartment B and boundary features. Right: Distribution of $-\log_{10}(p)$ (two-tailed binomial test, 5-kbp bin) for the depletion effect of siNIPBL and siCTCF on Rad21 ChIP-seq for each state (box plots with median and IQR). Source data are provided as a Source Data file.

relative interaction frequency and confirmed the contrasting depletion effects between cohesin/loaders and unloaders (Figs. 7e and S9b). siESCO1 showed a positive correlation with the unloaders, consistent with the role of ESCO1 with respect to facilitating loop stabilization and boundary formation[40], which is also supported by the loop length distribution (Fig. 3d). The stochasticity of cohesin pass-through could explain this stripe-like depletion effect at CTCF

roadblocks[25]. Considering that the effect is not symmetric and is one-sided, it would thus be more context specific.

## Depletion effects on long-range interactions
To further evaluate the correlation between epigenomic features and depletion effects on long-range interactions, we carried out peak-level and TAD-wide level comparisons within our Hi-C and ChIP-seq

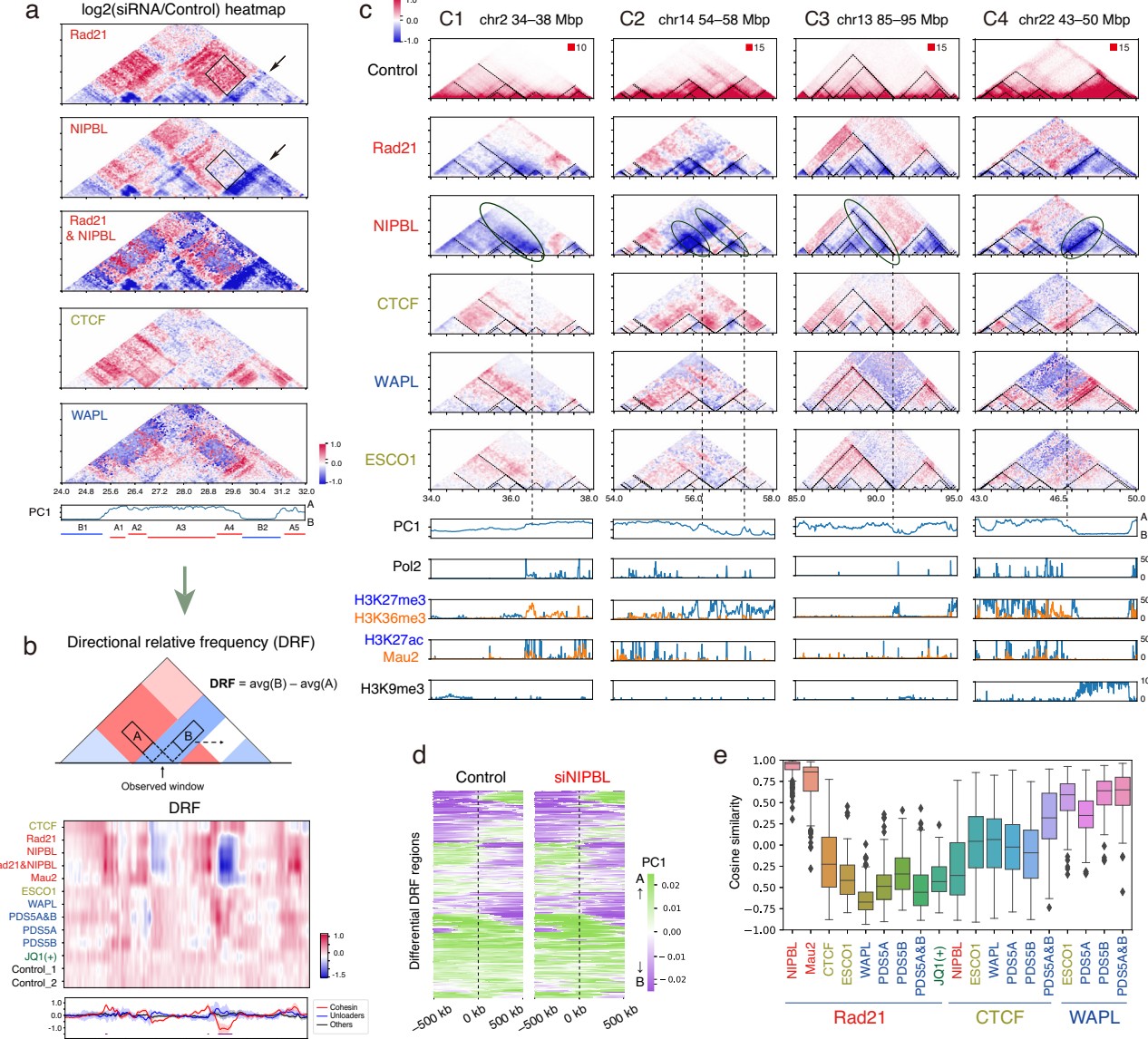

**Fig. 7 | InterTAD-level insulation is increased by cohesin loss. a** Relative enrichment of the interaction frequency (log scale) relative to control. The region and TAD labels (A1–A5, B1 and B2) are as in Fig. 4a. See Fig. S9a for all samples. **b** Top: Schematic illustration of DRF. Middle: Heatmap of DRF for depletions (row) for the same region as shown in (**a**). Bottom: Average DRF of cohesin and loaders (red), unloaders (blue), and others (black). The shaded regions indicate the 99% confidence interval. The purple lines at the bottom of the graph indicate the identified differential DRF regions. **c** Examples of differential DRF regions. Top: Normalized Hi-C matrix (control) and relative interaction frequency after depletions. The dashed black triangles indicate TADs in the control sample. Bottom: Compartmental PC1 values and ChIP-seq distribution. **d** Compartmental PC1 values for the control and siNIPBL samples centered on 241 differential DRF regions. Rows were ordered using hierarchical clustering based on the control sample. **e** The cosine similarity distribution of the relative interaction frequency between each Hi-C sample pair for all 241 differential DRF regions (-2 Mbp from the center of each region). Whisker plots with median and IQR. See Fig. S9b for all comparisons. Source data are provided as a Source Data file.

datasets. For the peak-level analysis, we applied a structured interaction matrix analysis (SIMA)[30,46] for a distance of 500 kbp–5 Mbp and did observe a striking difference among depletions (Figs. 8a and S10a). Whereas interactions between active markers (H3K4me2, H3K4me3, H3K27ac, Med1, and Pol2) increased after both siRad21 and siNIPBL, interactions between the suppressive marker H3K27me3 and the active markers decreased only after siNIPBL. CTCF depletion increased interactions, especially between promoter markers (Pol2, H3K4me2, and H3K4me3). Again, the depletion of cohesin unloaders showed the opposite tendency relative to siNIPBL. siESCO1 affected cohesin, CTCF, and enhancer markers. We found that the difference between siRad21 and siNIPBL was mainly derived from a global increase in long-range interactions (>2 Mbp) in siRad21, corresponding to interTAD

interactions (Figs. 8b and S10b). Although this increase could be involved in strengthened compartmentalization[17,18], it does not explain the difference between siRad21 and siNIPBL. Considering that longer-range interactions require more cohesin translocation from loading points to increase the probability of loop extension, our findings suggest that a different effect between the amount of cohesin on chromatin and the frequency of cohesin loading can appear in long-range interactions.

We also tested whether there is a region-wide depletion effect between TADs and the relationship to the epigenome. For this, we annotated all TADs with epigenomic marks and then classified all TAD pairs based on their relative change in interactions (k-means, $k = 5$; Fig. 9). There was a substantial decrease in interactions in clusters 2–4

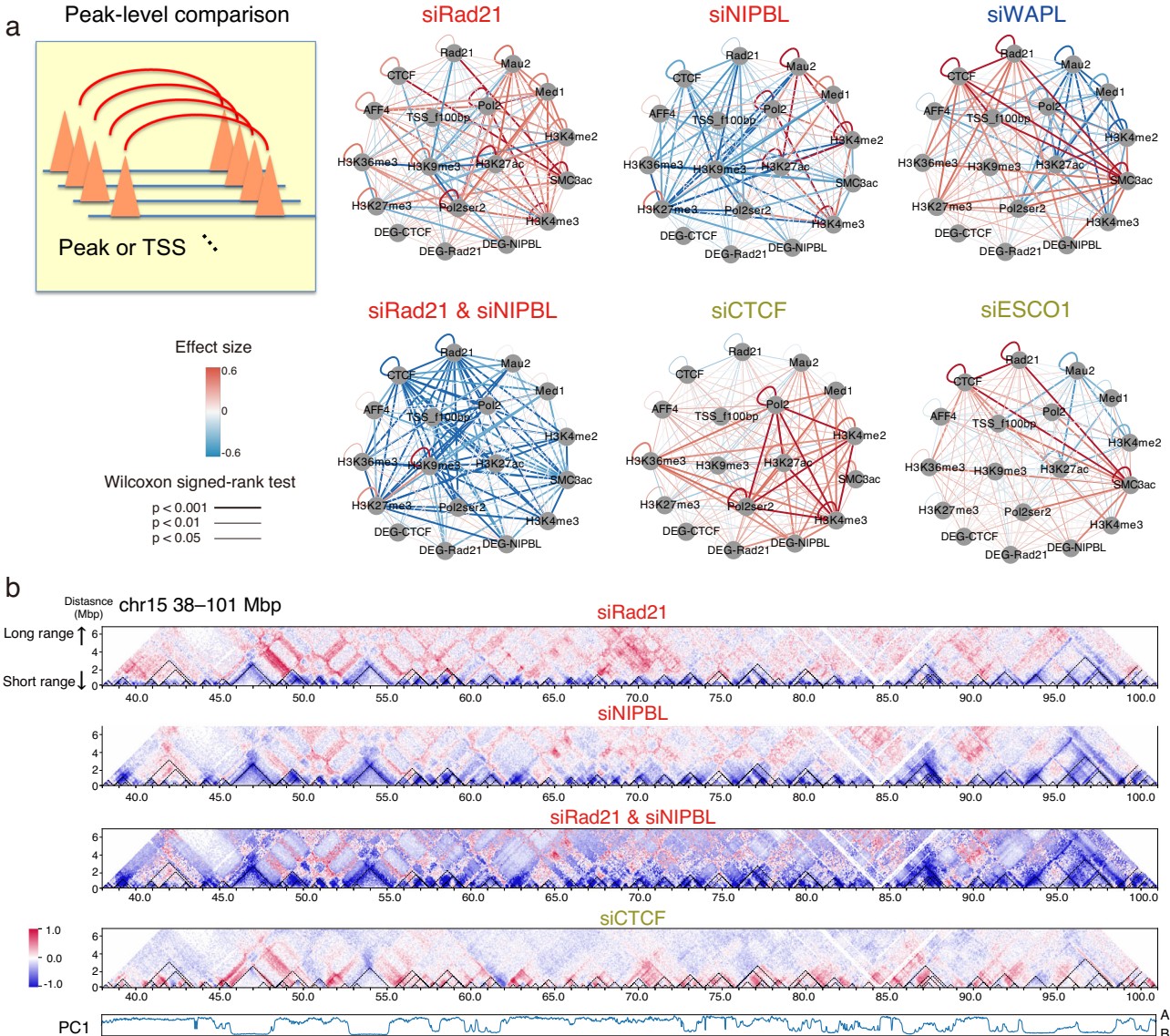

**Fig. 8 | A peek-level comparison of depletion effects on long-range interactions by SIMA analysis. a** Depletion effect on chromatin interaction (edges) between all pairwise features, including ChIP-seq peaks and the TSSs of DEGs (nodes) for six depletions as representatives. Edge color and width indicate the effect size and significance by the two-tailed Wilcoxon signed-rank test. See Fig. S10a for all samples. **b** Relative interaction frequency and compartmental PC1 values (chromosome 15, 38–101 Mb).

after the depletion of cohesin and loaders (Fig. 9b, c). These clusters were enriched for interactions between active and inactive markers (Fig. 9d), consistent with the depleted long-range interactions (A4–B2 in Fig. 7a). However, the effect is different between H3K9me3 and H3K27me3, for which the depletion effect in long-range interTAD interactions was more epigenomic dependent.

Lastly, we evaluated the correlation between DEGs and TAD-wide interTAD interactions. Although we found several genes for which changes in expression were consistent with a depletion effect on the interTAD interaction, the effect was not region-wide (e.g., *RUNX1* dysregulation was correlated but two neighboring DEGs, *KCNE1* and *DOP1B*, were not correlated; Fig. S11). Whether these DEGs result from TAD disruption or changes in interTAD interactions or if their expression is regulated independently via cohesin binding at TSSs or other gene-specific factors are important questions for future studies.

## Discussion

Despite multiple promising models[3,4,6,7,19], the cooperative and distinct roles of cohesin in combination with related factors concerning

chromatin folding and gene expression still need to be fully understood, especially in a context-specific manner. In this study, we generated a large-scale multi-omics dataset and developed a computational pipeline, named CustardPy, to systematically compare the effects of depleting cohesin and related factors. Although previous studies have often focused on the correlation between loop and TAD structures and gene expression, our methodology evaluates 3D features at various scales, including inter-TAD interactions. We found a variety of TAD boundaries and interTAD interactions that should be considered when investigating the functional and mechanistic relationships associated with cohesin. Whereas several studies have reported genome clustering using a single Hi-C sample—e.g., a third compartment[47] and subcompartment analysis[48,49]—our analysis compared multiple Hi-C samples and classified genomic regions based on the variation in depletion effects among samples (i.e., multiple-pair comparison in Fig. 1a). The perturbation of long-range interTAD interactions observed in this study cannot be captured by a typical analysis that evaluates only the number/strength of TADs and loops. A limitation of our approach is the cost associated with sample

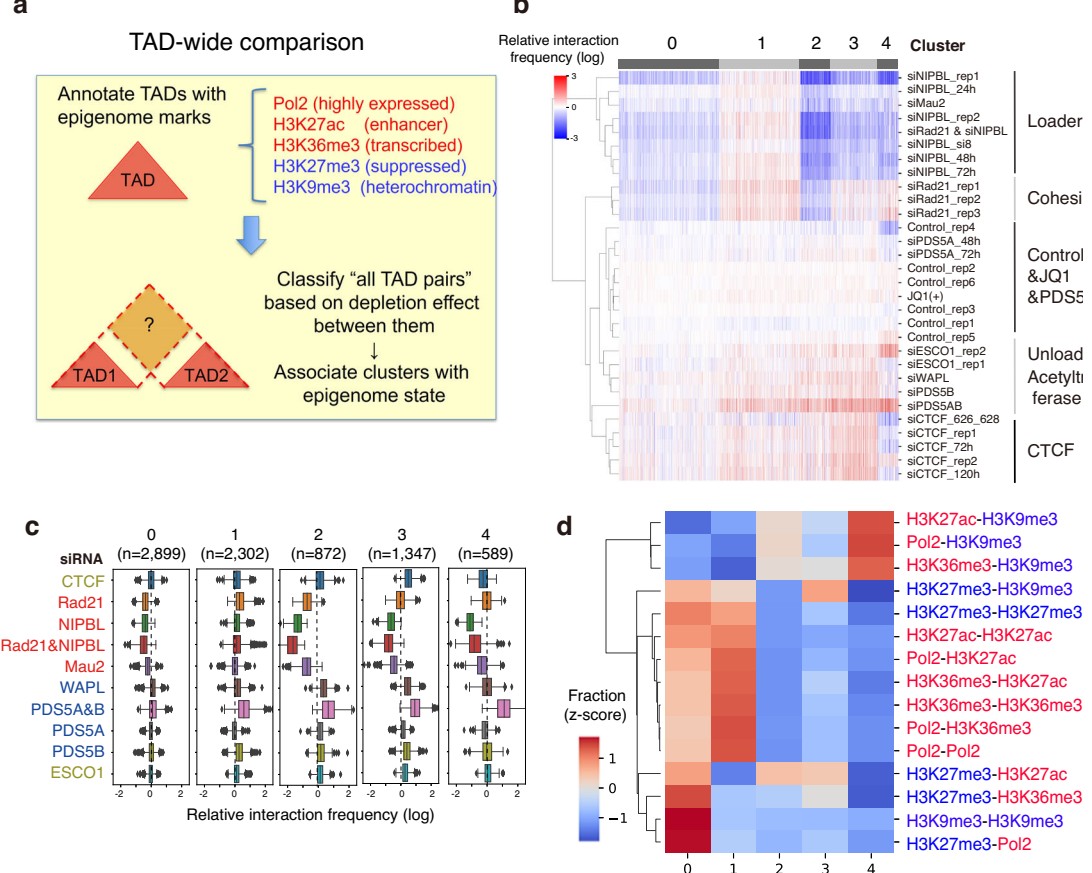

**Fig. 9 | The TAD-level comparison of depletion effects on interTAD interactions as compared with epigenomic marks. a** Schematic illustration of epigenome annotation of TADs. **b, c, d** A k-means clustering analysis ($k = 5$) of all TAD pairs based on the depletion effect on interactions between them (**b**), whisker plots with median and IQR of depletion effects on the clusters (**c**), and the fraction of the epigenomic state of TAD pairs included in the clusters (**d**). The numbers 0–4 indicate the cluster ids. Source data are provided as a Source Data file.

generation. Multi-sample pair comparisons require the generation of numerous samples, and long-range interaction analysis requires deep sequencing, both of which are costly. Another caveat is that the inter-TAD interaction analysis is dependent on the identified TADs, which varies between the tools used for TAD calling[50]. A future approach should enable tool-independent long-range interaction analysis to provide a more impartial perspective.

Most of the cohesin-related DEGs were related to the direct binding of cohesin around TSSs, which was more likely to be CTCF-independent. Some DEGs were also enriched near cohesin-separated boundaries, in which Mau2 specifically accumulated. In contrast, the disruption of TAD boundaries was not correlated with DEGs and histone modifications. Whereas the BRD4 mutation also causes a CdLS-like syndrome[32], there was little effect of JQ1 on chromatin folding despite the many isolated JQ1-related DEGs. Considering the report that BET inhibition does not disrupt enhancer–promoter contact[51], the CdLS phenotype might not involve the perturbation of chromatin structure but could be caused by direct transcription regulation by cohesin, e.g., by transcription machinery interacting with Pol2[7,41].

It should be noted that the siRNA system we used in this study has several limitations, such as incomplete depletion and potential secondary effects. For example, cohesin- and CTCF-independent boundaries may be lost after extreme depletion (e.g., with the auxin-inducible degradation system[17]). To circumvent these limitations, we focused here on the context-specific variation associated with the depletion of these factors by investigating the similarity and variability of the depletion effects. We delineated the

dominant factors for boundaries and interTAD interactions. In future experiments, this computational strategy should provide more insights as additional samples are analyzed using various conditions and other factors.

We demonstrated that cohesin is broadly distributed within compartment A, not only at peak sites of Rad21 or CTCF localization. Cohesin did not accumulate substantially in compartment B, where cohesin depletion had a minor effect. Because our siRNA system is independent of genomic context, this observation is unbiased and represents the distribution of cohesin abundance in the control sample. Under the loop extrusion model, cohesin distribution should have become more uniform after CTCF depletion because of the loss of cohesin stalling at CTCF sites. However, the unequal amounts of cohesin between compartments A and B were retained after siCTCF. In contrast, the interactions between neighboring TADs increased with the depletion of CTCF-dependent boundaries (e.g., Fig. 7a). Therefore, the amount of cohesin on the genome is not merely derived from loop extrusion but also is affected by the genomic context. How genomic segmentation within the whole genome is regulated by loop extrusion and other mechanisms remains an essential question for future studies.

## Methods

### Cell culture and siRNA
We used the siRNA system for depletion, as the auxin-inducible degradation system reduces protein levels even in the absence of auxin, which is unsuitable as a control[40]. RPE cells[3] were cultured in DMEM (Wako) supplemented with Penicillin-Streptomycin-L-

Glutamine Solution (Wako), 10% fetal bovine serum (Biosera), and 20 mM HEPES-KOH (pH 7.4). All siRNA transfections were performed using Lipofectamine RNAiMAX (Thermo Fisher Scientific) and the manufacturer's protocol 2 or 3 days before sample preparation, with a final RNA duplex concentration of 50 nM. The siRNA sequences are shown in Supplementary Data 7 and are the same as those described previously[28,52]. For inhibition of BET family proteins, cells were treated with JQ1 (Selleck Chemicals) for 6 h at a 1 μM final concentration. We labeled JQ1-treated and the corresponding control samples as JQ1(+) and JQ1(−), respectively.

### Antibodies

Antibodies used for ChIP and immunoblotting were as follows. Antibodies against histone H3 lysine-27 acetylation (H3K27ac)[53], H3 lysine-4 trimethylation (H3K4me3), H3 lysine-9 trimethylation (H3K9me3), H3 lysine-36 trimethylation (H3K36me3), and Pol2ser2 were provided by Dr. Kimura (Tokyo Institute of Technology, Tokyo, Japan). We also used antibodies against Rad21, Smc3ac, and ESCO1, which were described previously[52]. Antibodies against NIPBL (A301-779A, BETHYL), Mau2 (ab46906, abcam), CTCF (07-729, Merck), BRD4 (A301-985A50, BETHYL), AFF4 (A302-538A, BETHYL), Pol2 (14958, Cell signaling technology) and H3 lysine-27 trimethylation (H3K27me3, ab192985, abcam) were used for ChIP. Antibodies against NIPBL (sc-374625, Santa Cruz Biotechnology), α-tubulin (T6074, Merck), WAPL (16370-1-AP, Proteintech), PDS5A (A300-088A, BETHYL), PDS5B (ab70299, abcam), and CTCF (3417, Cell Signaling Technology) were used for immunoblotting. The mouse monoclonal antibody against Mau2 was generated using a synthetic peptide corresponding to residues 596–613 (PVQFQAQNGPNTSLASLL) of human Mau2 and used for immunoblotting. Antibody dilutions for immunoblotting were 1:500 (NIPBL and ESCO1) and 1:1000 (all other antibodies).

### Protein analysis

Cells were lysed with lysis buffer (20 mM HEPES-KOH, pH 7.5; 100 mM NaCl; 10 mM KCl; 10% glycerol; 340 mM sucrose; 1.5 mM $MgCl_2$; 10 mM sodium butyrate; 0.25% Triton X-100; 1 mM dithiothreitol; 1× cOmplete protease inhibitor cocktail [Roche]) as described[28]. The resulting lysate was mixed with SDS-PAGE sample buffer (50 mM Tris−HCl, pH 6.8; 2% SDS; 0.005% bromophenol blue; 7% glycerol; 5% 2-mercaptoethanol) and boiled for 5 min. The proteins were analyzed with a Mini-PROTEAN Tetra Vertical Electrophoresis Cell (Bio-Rad) following the manufacturer's protocol.

### In situ Hi-C

We used the in situ Hi-C protocol as described in Rao et al.[48]. In brief, ~3 × 10⁶ RPE cells were crosslinked with 1% formaldehyde for 10 min at room temperature, followed by an additional 5 min with 200 mM glycine in phosphate-buffered saline (PBS). Fixed cells were permeabilized in Hi-C lysis buffer (10 mM Tris−HCl, pH 8.0; 10 mM NaCl; 0.2% Igepal CA630; 1× protease inhibitor cocktail [Sigma]) on ice. The cells were treated with 100 U of MboI (New England Biolabs) for chromatin digestion, and the ends of digested fragments were labeled with biotinylated nucleotides followed by ligation. After DNA reverse crosslinking and purification, ligated DNA was sheared to a size of 300–500 bp using a Covaris S2 focused-ultrasonicator (settings: Duty Cycle, 10%; Intensity, 4; Cycles per Burst, 200; Duration, 55 s). The ligated junctions were then pulled down with Dynabeads MyOne Streptavidin T1 beads (Thermo Fisher Scientific). The pulled-down DNA was end-repaired, ligated to sequencing adaptors, amplified on beads, and purified using a Nextera Mate Pair Sample Preparation kit (Illumina) and Agencourt AMPure XP (Beckman Coulter). DNA was then sequenced to generate paired-end 150-bp reads using the Illumina HiSeq-2500 or X Ten system.

### Hi-C analysis by CustardPy

We developed various custom scripts for this study, most of which were integrated into the Hi-C analysis pipeline CustardPy (https://custardpy.readthedocs.io). It is written in Python3.7 and is available using the Docker system (https://www.docker.com/). CustardPy is designed to compare multiple Hi-C samples to evaluate the variation of depletion effects across multiple proteins (Fig. 1a), and all the tools used for Hi-C analysis described below are included in the latest version of the CustardPy docker image (version 1.2.0, https://hub.docker.com/r/rnakato/custardpy).

### Hi-C data processing with Juicer

Sequenced reads were processed using Juicer version 1.5.7 and Juicer tools version 1.9.9[54], with TADs and loops defined as in Rao et al.[48]. The detailed steps were as follows. Sequenced paired-end reads were mapped by BWA version 0.7.17[55] and converted to BAM format using Samtools v1.15 (http://www.htslib.org/). We then generated contact map files with square root vanilla coverage (VC_SQRT) normalization. We used 25-kbp resolution maps unless otherwise described. We called TADs using the Juicer tools *Arrowhead* command. Because the obtained TADs can be nested, we also generated a list of non-overlapping TADs by segmenting the genome based on all TAD boundaries. TAD boundaries were defined as edges for all annotated TADs. Loops were called at 5-kbp, 10-kbp, and 25-kbp resolution by the Juicer tools *HiCCUPS* command. To obtain peak-overlapping loops (Fig. 4c), we used BEDTools v2.28.0 (https://bedtools.readthedocs.io/en/latest/) and extracted loops for which both anchor sites overlapped with the peaks. High-resolution data that combined all replicates were generated by *mega.sh* script provided in Juicer. Eigenvector (PC1) values for compartment analysis were calculated by HiC1Dmetrics[44] because the *Eigenvector* command in Juicer tools sometimes failed. For allele-specific Hi-C analysis of chromosome X, we obtained single-nucleotide polymorphism data for RPE cells from Darrow et al.[56], which was then converted to genome build hg38 by the liftOver tool (https://genome-store.ucsc.edu/). We modified the *diploid.sh* script provided by Juicer and made interaction map files for active and inactive chromosome X.

The samples and mapping statistics are summarized in Supplementary Data 1. As we generated six replicates as control samples, we merged them into a single high-resolution Hi-C data. We used it to obtain reference data for the TADs, loops, and compartments. For the comparative analysis, we normalized Hi-C matrices based on the number of mapped reads on each chromosome. Therefore, the tendency for increases and decreases is relative; increased long-range interactions might be compensated for by increased short-range interactions[57]. siRad21 and siNIPBL were most affected by this fact, because almost all TADs and loops were depleted after these treatments. Therefore, our analysis focused on the variation of depletion effects across samples to capture the context-specific tendency, rather than translating the biological meaning of increased/decreased interactions.

### Hi-C data processing with other tools

To evaluate the quality and reproducibility of our Hi-C data, we used 3DChromatin_ReplicateQC[58], which internally implements QuASAR[59] and HiCRep[60]. Because of the large computational complexity involved, we used only chromosomes 21 and 22 with a 50-kbp bin for the quality evaluation. We confirmed that all of our Hi-C data had sufficient quality (QuASAR-QC scores > 0.05; Supplementary Data 1). HiCRep was used to evaluate the overall similarity of the depletion effects among our Hi-C samples by calculating a stratum-adjusted correlation coefficient that captures the similarity of chromatin features including TADs and loops. We used Cooler[61] and cooltools (https://cooltools.readthedocs.io/) for APA plots, plots of average TAD data, and saddle plots. We used GENOVA[62] to calculate the

compartment strength. The Hi-C matrices with ChIP-seq distributions were visualized using CustardPy commands.

## Structured interaction matrix analysis (SIMA)

To explore interactions between specific chromatin features (e.g., ChIP-seq peaks), we used a SIMA[46] implemented in HOMER (http://homer.ucsd.edu/homer/). SIMA assembles information for multiple occurrences of each feature, providing an overview of Hi-C interactions associated with a genomic feature between each pair of specified domains. In this study, the genomic features included the ChIP-seq peak list (AFF4, CTCF, H3K27ac, H3K27me3, H3K36me3, H3K4me3, H3K4me2, Smc3ac, Mau2, Med1, Pol2, Pol2ser2, and Rad21), and TSSs of DEGs (siCTCF, siNIPBL, siRad21). Domains of interest were defined as TAD lists, and the distance between two TADs was specified to be <5 Mbp, <2 Mbp, or 2 to <5 Mbp with '-max -min' parameters. By comparing the background model, we obtained an enrichment score, representing the degree to which a genomic feature pair was enriched in the Hi-C interactions between two TADs. To compare differences in enrichment scores between cohesin-knockdown and control Hi-C samples, we used the paired Wilcoxon signed-rank test to calculate the p-value and effect size for each genomic feature pair, as described[30]. We used Cytoscape v3.8.2 (https://cytoscape.org/) to visualize the results.

## Multi-scale insulation score

CustardPy calculates a multi-scale insulation score as described[42]. In brief, the insulation score was calculated at a resolution of 25 kbp as the log-scaled relative contact frequency across pairs of genomic loci located around the genomic positions from 100 kbp to 1 Mbp. The 500-kbp distance was used in the single insulation score analysis. For the classification of insulation boundaries into six types, we used the following criteria based on this 500-kbp insulation score:

1. if(siNIPBL − control) > $T_{ins}$ and if(siCTCF − control) > $T_{ins}$: "all-dependent"
2. else if(siNIPBL − control) > $T_{ins}$ or if(siRad21 − control) > $T_{ins}$: "cohesin-dependent"
3. else if(siCTCF − control) > $T_{ins}$: "CTCF-dependent"
4. else if(control − siCTCF) > $T_{ins}$: "CTCF-separated"
5. else if(control − siNIPBL) > $T_{ins}$ or if(control − siRad21) > $T_{ins}$: "cohesin-separated"
6. else if(siNIPBL − control) < $T_{ins}$ and if(siCTCF − control) < $T_{ins}$ and if(siCTCF − control) < $T_{ins}$: "robust"

where we set $T_{ins}$, the threshold of the insulation score, as 0.13. We excluded siMau2 as a criterion because it had a smaller effect than siNIPBL and siRad21 on the insulation score. We excluded chromosomes X and Y and the mitochondrial genome from this boundary analysis. The obtained six boundary types are summarized in Supplementary Data 5.

## Directional relative frequency (DRF)

CustardPy can calculate the DRF that identifies the directional bias of depletions on interTAD interactions[44]. DRF measures the bias in the relative interaction frequency $M = \log(C_{siRNA}) - \log(C_{control})$ between regions up- and downstream of each genomic region, where $C$ is a normalized contact matrix. Therefore, the DRF can be calculated by

$$\text{DRF}_i = \sum_{j=l_{min}}^{l_{max}} M_{i,i+j} - \sum_{j=l_{min}}^{l_{max}} M_{i,i-j}, \qquad (1)$$

Where $l_{min}$ and $l_{max}$ indicate the range of the interaction. In this study, we set $l_{min} = 500$ kbp and $l_{max} = 2$ Mbp.

To obtain differential DRF regions, we classified Hi-C samples as "cohesin and loaders," "cohesin unloaders," and "others (including control)". We then calculated the average DRF values and 99% confidence intervals (CIs) for each region. We used these values to identify the regions that satisfied the following criteria: the 99% CI ranges of "cohesin and loaders" and "others" did not overlap, and the average DRF value of "cohesin and loaders" was > $T_{DRF}$ or <−$T_{DRF}$, where $T_{DRF}$ refers to the DRF threshold. We set $T_{DRF} = 0.7$ in this study. The obtained differential DRF regions are summarized in Supplementary Data 6.

## RNA-seq

Total RNA was isolated using Trizol (Thermo Fisher Scientific) and a Nucleospin RNA kit (Macherey-Nagel). rRNA was removed with the Ribo-Zero Gold rRNA Removal kit (Illumina), followed by sequencing library preparation with the NEBNext Ultra Directional RNA Library Prep kit for Illumina (New England Biolabs). Single-end 65-bp reads were sequenced by the Illumina HiSeq-2500 system. Sequenced reads were mapped to the human reference sequence (GRCh38) by STAR version 2.7.3a[63] with the following options "SortedByCoordinate --quantMode TranscriptomeSAM --outSAMattributes All". The samples and mapping statistics are summarized in Supplementary Data 2. The gene expression levels were estimated by RSEM version 1.3.1[64] with the option "--estimate-rspd --strandedness reverse". We used DESeq2[65] to identify DEGs (protein-coding genes, false discovery rate [FDR] <0.01). We focused on protein-coding genes to avoid the effects of repetitive non-coding RNAs.

To mitigate the indirect effect and the technical variances, we generated the list of DEGs by merging the top-ranked 1000 DEGs from each pairwise comparison between each siRNA and the controls. For clustering RNA-seq samples based on DEGs, we used the Simpson index to compare the binary overlap of DEGs. We did not adopt quantitative clustering using the z-score of gene expression level because we prepared the RNA-seq samples by three different experiments, each with control samples, and the quantitative analysis would be more affected by the technical variations due to sample preparation. For clustering DEGs, we generated DEG vectors of each depletion containing {1: upregulated, −1: downregulated, 0: not included} and created a matrix by combining the vectors of all depletions. We applied K-means clustering to the matrix and classified the DEGs into 20 clusters. We used clusterProfiler[66] for the GO enrichment analysis.

## Spike-in ChIP-seq

Chromatin preparation for ChIP was performed as described[7]. In brief, ~8 × 10^6 RPE cells were crosslinked with 1% formaldehyde for 10 min at room temperature, followed by an additional 5 min with glycine in PBS added at a final concentration of 125 mM. Fixed cells were lysed in LB1 (50 mM HEPES-KOH, pH 7.4; 140 mM NaCl; 1 mM EDTA; 10% glycerol; 0.5% NP-40; 0.25% Triton X-100; 10 mM dithiothreitol; 1 mM PMSF) on ice. The lysate was washed sequentially with LB2 (20 mM Tris-HCl, pH 8.0; 200 mM NaCl; 1 mM EDTA; 0.5 mM EGTA; 1 mM PMSF) and LB3 (20 mM Tris-HCl, pH 7.5; 150 mM NaCl; 1 mM EDTA; 0.5 mM EGTA; 1% Triton X-100; 0.1% sodium deoxycholate; 0.1% SDS; 1× cOmplete protease inhibitor cocktail [Roche]) on ice. The lysate was resuspended in LB3 and sonicated using a Branson Sonifier 250D (Branson) for chromatin shearing (12 sec with amplitude set at 17% of the maximum amplitude, six times). In addition, lysate containing fragmented chromatin was also prepared from ~2 × 10^6 mouse cells (C2C12) with the same procedures. Human cell lysate and mouse cell lysate (as a spike-in internal control) were combined (~4:1 ratio) and incubated with protein A or G Dynabeads (Thermo Fisher Scientific) conjugated with 2 μg of the relevant antibodies for 14 h at 4 °C. The beads were then washed five times with cold RIPA wash buffer (50 mM HEPES-KOH, pH 7.4; 500 mM LiCl; 1 mM EDTA; 0.5% sodium deoxycholate; 1% NP-40) and once with cold TE50 (50 mM Tris-HCl, pH 8.0; 10 mM EDTA). Material captured on the beads was eluted with TE50 containing 1% SDS. The eluted material and input were incubated for 6 h at 65 °C to reverse crosslinks. They were then treated with 100 ng RNaseA (Roche) for 1 h at 50 °C, followed by treatment with 100 ng Proteinase K (Merck)

overnight at 50 °C. The input and ChIP DNA were then purified with a PCR purification kit (Qiagen). DNA from the ChIP and input fractions was end-repaired, ligated to sequencing adaptors, amplified, and size-selected using NEBNext Ultra II DNA Library Prep kit for Illumina (New England Biolabs) and Agencourt AMPure XP (Beckman Coulter). DNA was then sequenced to generate single-end 65-bp reads using the Illumina HiSeq-2500 and NextSeq 2000 systems.

Reads were aligned to the human genome build hg38 and mouse genome build mm10 using Bowtie2 version 2.4.1[67] with default parameters. Quality assessment was performed with SSP version 1.2.2[68] and DROMPAplus version 1.12.1[69]. Spike-in read normalization, peak calling, and visualization were performed with DROMPAplus. The mapping statistics, quality values, and scaling factors for spike-in normalization are summarized in Supplementary Data 3. The default parameter set was used for peak calling (100-bp bin, --pthre_internal 5, --pthre_enrich 4) except for H3K9me3 (--pthre_internal 1 --pthre_enrich 2) due to the lower signal-to-noise ratio. For read visualization, we displayed $-\log_{10}(p)$ scores of ChIP/input enrichment (--showpenrich 1 option), which is recommended for distinguishing the signal from the noise[70].

### Permutation test for overlapping analysis

To compare the overlapping ratio of TSSs of DEGs and ChIP-seq peaks, Hi-C loops, and insulation boundaries, we implemented a permutation test ($n = 1000$) that compared the relative overlap frequency against the background distribution. As a background, we used all DEGs obtained by DESeq2 (11,345 genes, FDR < 0.01) for DEGs and all boundaries (7421) for the six types of boundaries. We randomly picked up the same number of genes or boundaries from the background in each permutation and generated the frequency distribution. For the boundary analysis, we counted DEGs and the peaks that overlapped within 50 kbp of them.

### Correlation of interactions with epigenomes

For interTAD interaction comparisons, we extracted all TAD regions with widths of >100 kbp and annotated them using the epigenomic marks (H3K36me3, H3K27me3, H3K9me3, and Pol2) if the marks covered >40.0% of the TAD length. To avoid a low read coverage at long-range distances and the technical effect derived from the different resolutions of Hi-C matrices, we used the log-fold change $\log_2(N_{siRNA}/N_{control})$ where $N_{siRNA}$ and $N_{control}$ indicate the total number of fragments mapped within the interTAD regions between a TAD pair (with ≤2 Mbp between them) annotated with the epigenomic marks. We calculated scores for each TAD pair and applied k-means clustering ($k = 5$). Then we calculated the z-score–normalized fraction of epigenomic status for the TAD pairs included in each cluster to estimate the epigenomic-dependent depletion effect of interTAD interactions.

### Extended ChromHMM

Our previous study showed that several one-dimensional metrics for Hi-C data are effective for annotating chromatin states in detail[44]. In this study, we added CTCF, TAD boundaries, and compartment information in addition to five core histone marks (H3K4me3, K3K27ac, H3K27me3, H3K36me3, and H3K9me3) to ChromHMM and annotated 15 chromatin states.

### Reporting summary

Further information on research design is available in the Nature Portfolio Reporting Summary linked to this article.

## Data availability

The human reference genome hg38 was obtained from the UCSC Genome Browser (https://genome.ucsc.edu/). The raw sequencing data and processed files for the Hi-C, RNA-seq, and ChIP-seq data from this study have been submitted to the Gene Expression Omnibus (GEO) under the accession number GSE196450. The .hic files of the merged Hi-C samples and the reference TAD and loop files are also available on GSE196034. The reference data of TAD and loops obtained from the merged control sample are available on Zenodo (https://doi.org/10.5281/zenodo.8218447). Source data are provided with this paper.

## Code availability

The original code used for the principal analysis is available on Zenodo (https://doi.org/10.5281/zenodo.8218447). The CustardPy docker image is available on DockerHub (https://hub.docker.com/r/rnakato/custardpy).

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

## Acknowledgements

We thank all members of the Nakato and Shirahige Laboratories for their discussions and comments on the manuscript. This work was supported by a Grant-in-Aid for Scientific Research (17H06331 and 23H02466 to R.N. and 20H05686 and 20H05940 to K.S.), the Japan Agency for Medical Research and Development under grant number JP23gm6310012h0004 and the Japan Science and Technology Agency under grant number JPMJCR18S5.

## Author contributions

R.N. conceived this project and wrote the manuscript. R.N., J.W., L.A.E.N., and G.M.O. implemented the computational analysis. R.N. and Y.N. developed and tested CustardPy. T.S. prepared Hi-C, ChIP-seq, and RNA-seq samples. M.B. designed ChIP-seq and RNA-seq samples. K.S. supervised the sample preparation and sequencing and suggested ways to improve the analysis and the manuscript.

## Competing interests

The authors declare no competing interests.
