## [Peer Review File · Nature Communications]

Context-dependent perturbations in chromatin folding and the transcriptome by cohesin and related factorsREVIEWER COMMENTS

Reviewer #1 (Remarks to the Author):

This study presents an in-depth and integrative analysis of chromatin structure changes, gene expression changes as well as genomic binding sites after depletion of the cohesin complex and its regulators (loaders, unloaders and CTCF) in RPE1 cells. For this, the authors have generated a huge amount of data with different NGS approaches (CHIP-seq, RNA-seq, Hi-C).

The in-depth analysis of these data confirms previous observations obtained for individual protein depletions of one or the other protein in other cells types. Therefore the strength and impact of the study lies in the integration of all those data generated in the same cell type.

To the opinion of this reviewer, this study provides important resources for the field. However, the manuscript needs to be very much strengthened to be suitable for publication.

A major product of this study is the CustardPy data analysis pipeline, as highlighted by the authors in the abstract. However, a description of the pipeline and information which analyses were actually generated by the pipeline is completely missing. Also, a discussion on strengths and limitations of the pipeline is not there.

Most figure panels present very complex analyses involving multiple datasets and sometimes different analyses in parallel. This complexity obstructs the appreciation of very interesting observations. The authors claim to have made three major observations: "1) Gene expression dysregulation that was correlated with splitting of TADs (i.e., TAD splits) was associated with the loss of cohesin. 2) There was an imbalanced enrichment of cohesin binding on chromatin between active and inactive chromosome regions, which persisted even after CTCF depletion. 3) Perturbation of long-range interactions was correlated with epigenomic states of TADs.". This reviewer would suggest to reorganize/simplify figures so that the data behind these statements can be clearly appreciated. In particular observations that would not be possible without implementing all the different data types in this pipeline should be highlighted.

Specific points for the revision:

The manuscript would benefit from a thorough textual revision. One example, in line 144 the authors state that "depletion of MAU2 leads to highly depleted long loops". This could be rephrased into "depletion of MAU2 leads to strong reduction of long loops".

Fig. 1: It is unclear which data were clustered in panel D. Also, the relevance of the figure is not clear. Eventually the full figure could be better placed into the supplement and a reduced version showing one replicate of each depletion in panel 1D. Also, the figure shows a quite interesting behavior of cohesin peaks with SMC3 acetylation. The authors should consider to discuss this – why are SMC3ac sites reduced in specific clusters?

Fig. 1E: If this reviewer understands this figure correctly, the deletion of RAD21 by siRNA would only lead to a rather moderate reduction of RAD21 binding sites?

Fig. 3: The headline indicates epigenome data, but there are no epigenetic data in the figure.

Fig. 3C: What is the meaning of the dots within the boxes?

Fig. S1 Why are the western blots from Fig. 1C shown again?

Depletion of the different cohesin subunits and regulator leads to a change in the cell cycle distribution. For example, PDS5A and PDS5B depletion seem to strongly enrich cells in G1-phase. How does this impact the data generated with the different approaches as well as the data analysis?

Fig. 7A The relevance of this figure is unclear. The figure legend is not informative.

line 110 – the authors evaluate similarity of the Hi-C samples. What are the criteria for this analysis?

Line 117 – the figure reference is not correct

Line 131-132 – unclear whether the claim is really that TAD formation is strengthened by the depletion of cohesin and loaders

Line 366 – The authors refer to other studies using only single Hi-C samples. It is unclear what the authors intend to say - a single replicate or depletion of only one factor?

Reviewer #2 (Remarks to the Author):

Overview:

The article presents a comparative analysis of the effects of the knockdown of cohesin or related factors on chromatin folding, transcriptome, and histone modifications in human RPE cells. The authors employed a combination of in situ Hi-C, spike-in ChIP-seq, and RNA-seq data to investigate the impact of siRNA-depleted cellular context on these molecular mechanisms. The study initially examined the separate effects of factor depletion on TADs and loops, compartments, and DEGs using Hi-C, RNA-seq, and ChIP-seq data. Subsequently, the authors integrated these different patterns to analyze the effects on chromatin folding and transcriptome in a context-dependent manner. Notably, the author found a substantial depletion of interTAD interactions between "active TAD" and "inactive TAD", which was region-specific and asymmetrical. These findings align with previous research while also presenting novel insights.

The authors' presentation of their findings through the use of figures and their comprehensive analysis are noteworthy. However, there remain several areas that would benefit from further elaboration and clarification.

Major:

"The comparative Hi-C analysis":

1. it is observed from Figures 2B and 2C that the co-depletion of PDS5A and PDS5B results in a mild decrease in compartment strength (~10 Mbp). The description of the result and its possible reason could be further elaborated upon to enhance the novelty of the work.
2. Page 7, line 141, the statement "After siRad21, most short loops were depleted, and the distribution peaked at a longer length (~10 Mbp) than the control (~500 kbp) ..." appears to be inconsistent with the data presented in Figure 2D. In this figure, the control peaked at approximately 400 kbp, while siRad21 only exhibited a slightly higher frequency around 800 kbp loops after losing short loops. Additionally, siCTCF displayed a peak at 500 kbp and the NIPBL-Rad21 co-depletion was less dramatic than siRad21. Therefore, it is recommended that the entire paragraph be reviewed and corrected.

"The comparative RNA-seq analysis":

1. The correlation map presented in Figure 3A employs the Simpson index, which solely considers the number of overlapping DEGs across siRNA samples and disregards their expression levels. This raises the concern of introducing bias in the correlation map, as it treats up- and down-regulated DEGs equally. Therefore, it would be beneficial to address this issue and consider alternative methods that account for differential expression levels.
2. Regarding Figure 3B, the description of the K-means clustering of DEGs is rather vague, and it would be helpful to clarify the input data and distance calculation method employed.
3. In Figure 3C, the observed increase in the enrichment of Rad21 peaks in the siRad21 sample, particularly in clusters 0, 2, 5, 7, 10, 14, and 18, raises questions regarding the underlying cause. Therefore, it would be beneficial to discuss potential explanations for this phenomenon.
4. On Page 9, line 195, the statement "Acetylated cohesin ... than non-acetylated sites (Figure 3C)" lacks clarity and would benefit from a more detailed explanation and an accompanying illustration.
5. Figure 3D displays the top-ranked DEGs, and it would be informative to know whether they are up- or down-regulated after depletion and the potential reasons for such changes.
6. Figure 3E illustrates that siCTCF significantly increases cohesin peaks at upstream and exon. Please provide possible explanations.

"The multi-scale insulation of TAD boundaries":

1. In Figure 4D, the insulation score of TAD boundaries increased after depletion but was labeled as "Boundary loss (TAD fusion)". I wonder if it's wrong as the boundary appears to be strengthened.
2. On page 10, line 231, "resulting in the dysregulation of the expression of genes in the surrounding region". However, the current evidence does not establish a clear relationship between the cohesin-separated boundaries and the position of the DEGs.
3. On page 10, line 233, "would be more highly correlated with DEGs than ...". It may be more appropriate to state that the insulation score is more highly correlated with "up-regulated DEGs".

"The cohesin distribution over AB compartment":

1. On page 12, line 263, the author suggests that "there are different amounts of cohesin between compartment A and B". which could be quantified and compared using Rad21 ChIP-seq of the control. I wonder if the amount difference of depleted cohesin between compartment A and B is due to the depletion bias in the siRNA system. If so, the statement could not reflect the amounts of cohesin in the control and should be changed to "different amount of affected cohesin". As the same on page 12, line 269 "the cohesin accumulated to the highest levels at ...", it might need to change to "the depleted cohesin accumulated to ...". Furthermore, I wonder if the biased affection on the A/B compartment is observed from the whole genome or just the illustrated part in Figure 5A, chr21: 24-32 Mbp?
2. On page 12 Line 266, the author states that "a milder loss of intraTAD interactions in Strong B than in Strong A regions, whereas siCTCF affected both." However, it appears that there was a more severe loss of intraTAD interactions in the Strong B region in Figure 5C. The author should clarify and correct this consistency.

"Methods":

On page 2, line 23, page 4, line 77, and page 16, line 360, the author mentioned the main component of their comparative method, CustardPy. However, the Methods section lacks a description of the function or models employed by CustardPy. An overview of CustardPy is deemed necessary.

Minor:

1. Please explain the statement on Page 3, line 57 "... suggesting that CTCF boundaries are not absolute."
2. On page 3, line 60, "Cohesin turnover is also critical for proper gene expression for regulation and for the CdLS phenotype". Adding some examples will be much helpful.
3. Page 3, line 67, "Moreover, cohesin and CTCF also localize within TADs without forming boundaries." Please provide its reference.
4. Page 9, line 203, "other regions" can be replaced by "BMP6's exon, SSBP2's intron, and the intergenic region".
5. The image resolution of Figure 5B (left) is not sufficient to show the thresholds clearly that separate the strong A region and weak A region.

Reviewer #3 (Remarks to the Author):

The manuscript "A comparative multi-omics analysis of context-dependent perturbations in chromatin folding and the transcriptome by cohesin and related factors" by Nakato et al. introduces a large multi-omics dataset of cohesin perturbation and a computational workflow, CustardPy, for multi-omics (replicated Hi-C, RNA-seq, spike-in ChIP-seq) data analysis (Python3 and two Docker images implementation, with documentation). The introduction briefly introduces the role of cohesin and the associated factors (loaders, unloaders, CTCF, and others). The method is applied to define similarities and differences in the effect of depletion of individual factors by siRNAs in human retinal pigment cells. Besides validating previous observations, the authors report the association of gene expression changes with TAD splitting, imbalanced enrichment of cohesin binding in A and B compartments (A1-5, B1, B2), the differential effect of cohesin and loaders depletion on short- and long-range interactions within TADs, differential effects of CTCF and cohesin. Investigation of the allele-specific depletion effect on chromosome X. The authors use creative methods for data analysis, e.g., considering "Weak" and "strong" A/B compartments, directional relative frequency, classification of insulation boundaries into six subtypes. There are also novel methods for visualizing the multi-omics results, e.g., Figure S5B showing the proximity of DEGs to disrupted TAD boundaries. All datasets are well-documented, and all main and supplementary figures and tables are well-described. Methods used for data analyses are considered gold-standard (e.g., Juicer, Cooler for Hi-C, DESeq2 for RNA-seq). The CustardPy method is implemented in XYZ and wrapped in a Docker image.

Minor

- The data is a big part of the submission. Suggest renaming files on GEO for better compliance with the FAIR principles. Currently, the file names are very heterogeneous, and hard to understand which one belongs to which technology/condition. Systematic file naming would facilitate finding the right data.
- The Custardpy documentation describes both Docker and Singularity installations, However, the following usage examples are for Singularity only. It may be helpful to provide Docker syntax as the primary examples, supplemented with Singularity examples (as tabs or expandable sections).
- "Eigenvector (PC1) values for compartment analysis were calculated by HiC1Dmetrics (because the Eigenvector command in Juicer tools." - seems unfinished, and the HiC1Dmetrics is unreferenced.
- Saddle plots are often difficult to quantify. Suggesting to use an approach used in Du et al., "DNA Methylation Is Required to Maintain Both DNA Replication Timing Precision and 3D Genome Organization Integrity.", Figure 4B-F (they provide code).
- Stripe-like structures were detected. Tools started to appear to analyze them (FIREcaller, CHES). It may be a good idea to use them for the sake of standardizing analyses. And, include it in the Docker images.

Major changes in the revised manuscript:

- We have added detailed information about our new tool CustardPy. We have added the figure showing its workflow and explanations of its advantages and limitations in the Introduction and Discussion sections.
- We have reorganized the main figures and main results to make the manuscript easier to understand.
- We have added several experiments suggested by the reviewers.
- We have updated the file names in the GEO database to make them clearer.
- Followed the FORMATTING INSTRUCTIONS of Nature Communications: Shortened the Title and the subheadings of Results section; lowercased the panel names of the figures; added the Code Availability section; uploaded our custom scripts and data to Zenodo instead of GitHub.

In addition, we also released a major release of CustardPy (version 1, <https://github.com/makato/CustardPy>), which includes many new features used in this project, and dramatically improved the explanation on the manual website. We hope that the explanations and revisions to our paper are satisfactory.

Reviewer #1:

This study presents an in-depth and integrative analysis of chromatin structure changes, gene expression changes as well as genomic binding sites after depletion of the cohesin complex and its regulators (loaders, unloaders and CTCF) in RPE1 cells. For this, the authors have generated a huge amount of data with different NGS approaches (CHIP-seq, RNA-seq, Hi-C).

The in-depth analysis of these data confirms previous observations obtained for individual protein depletions of one or the other protein in other cells types. Therefore the strength and impact of the study lies in the integration of all those data generated in the same cell type.

To the opinion of this reviewer, this study provides important resources for the field. However, the manuscript needs to be very much strengthened to be suitable for publication.

We thank the reviewer for the consideration of our manuscript, and detailed feedback on our work.

A major product of this study is the CustardPy data analysis pipeline, as highlighted by the authors in the abstract. However, a description of the pipeline and information which analyses were actually generated by the pipeline is completely missing. Also, a discussion on strengths and limitations of the pipeline is not there.

We thank for the reviewer's pointing out this. To clarify the analyses done by CustardPy, we added a new figure of CustardPy workflow (Figure 1a) the explanation of it in Introduction section (lines 82-85). We also added an explanation about the strengths and limitations of CustardPy in Discussion section (lines 388-399).

Most figure panels present very complex analyses involving multiple datasets and sometimes different analyses in parallel. This complexity obstructs the appreciation of very interesting observations. The authors claim to have made three major observations:

“1) Gene expression dysregulation that was correlated with splitting of TADs (i.e., TAD splits) was associated with the loss of cohesin.

2) There was an imbalanced enrichment of cohesin binding on chromatin between active and inactive chromosome regions, which persisted even after CTCF depletion.

3) Perturbation of long-range interactions was correlated with epigenomic states of TADs.”.

This reviewer would suggest to reorganize/simplify figures so that the data behind these statements can be clearly appreciated.

We apologize for the inconvenience for the complexity of our explanation. In fact, in our study, we performed various analyses using multi-omics data to investigate the perturbation pattern from multiple angles, which would make the results section complex. To better illustrate the correspondence between the results and the figures, in the revised manuscript we have split the third of the main observations into two and quoted the corresponding figures as follows (lines 89-95):

- 1) Gene expression dysregulation that was correlated with splitting of TADs (i.e., TAD splits) was associated with the loss of cohesin (Figure 5).
- 2) There was an imbalanced enrichment of cohesin binding on chromatin between active and inactive chromosome regions, which persisted even after CTCF depletion (Figure 6).
- 3) CustardPy identified the context-specific pattern of inter-TAD interactions between depletions (Figure 7).
- 4) Perturbation of long-range interactions was correlated with epigenomic states of loop anchors

and TADs (Figures 8 and 9).

In particular observations that would not be possible without implementing all the different data types in this pipeline should be highlighted.

We have outlined what was done by CustardPy in the workflow (Figure 1a) and the Discussion section (lines 388-392). While other tools use a single Hi-C sample or a pair of samples to classify genomic regions, CustardPy compares multiple Hi-C samples to assess the variation of depletion effects across multiple proteins and uses this for genome classification. Therefore, Figures 5, 7, and 9 would not be possible without CustardPy.

Specific points for the revision:

The manuscript would benefit from a thorough textual revision. One example, in line 144 the authors state that “depletion of MAU2 leads to highly depleted long loops”. This could be rephrased into “depletion of MAU2 leads to strong reduction of long loops”.

Thank you for the comment. In the original manuscript we have submitted, the sentence was “siMau2 resulted in highly depleted long loops (~1 Mbp), and the distribution peaked at a shorter length than the control (~400 kbp).” We are afraid that the wrong version of the manuscript was sent to you by mistake. We hope that the explanations in the revised manuscript are more readable and satisfactory.

Fig. 1: It is unclear which data were clustered in panel D. Also, the relevance of the figure is not clear.

Figure 1D in the original manuscript (Figure 2c in the revised manuscript) describes the clustering results of our all Hi-C samples based on the similarity score “a stratum-adjusted correlation coefficient” calculated by HiCRep. We added the explanation in the figure legend.

Eventually the full figure could be better placed into the supplement and a reduced version showing one replicate of each depletion in panel 1D.

This clustering analysis serves two purposes. The first is to explore the similarity and variance of the depletion effects on the Hi-C data using hierarchical clustering. The second is to confirm that there is sufficient similarity between replicates of each depletion. Therefore, the inclusion of all replicates is important. We ask for your understanding in this matter. We have added a sentence

to clarify this point (lines 119-120).

Also, the figure shows a quite interesting behavior of cohesin peaks with SMC3 acetylation. The authors should consider to discuss this – why are SMC3ac sites reduced in specific clusters?

Since Figure 1D in the original manuscript describes the clustering results of Hi-C samples, it does not contain the information of the SMC3 acetylation peaks. We are afraid we are not clear about which panel the review is referring to.

Fig. 1E: If this reviewer understands this figure correctly, the deletion of RAD21 by siRNA would only lead to a rather moderate reduction of RAD21 binding sites?

Since Figure 1 in the original manuscript does not include panel E, we are not sure which panel the reviewer is referring to. To answer in a general sense, because the depletion efficiency of siRNA is not perfect, the remaining cohesin still binds to chromatin, which is ChIPed and amplified by PCR in ChIP-seq. This would be the reason why the Rad21 ChIP-seq data of siRad21 looks moderate in effect.

Fig. 3: The headline indicates epigenome data, but there are no epigenetic data in the figure.

In Figure 3 in the original manuscript (Figure 4 in the revised manuscript), panel D shows the enrichment pattern of the ChIP-seq data, and panels C and E show the results of the comparison of DEGs and ChIP-seq peaks. We would say that these are the epigenomics data.

Fig. 3C: What is the meaning of the dots within the boxes?

We apologize for the inadequate explanation. In Figure 3C in the original manuscript (Figure 4c in the revised manuscript), the dots indicate the significant enrichment or loss of ChIP-seq peaks (left) and Hi-C loops (right) at the TSSs of DEGs in the clusters compared to all DEGs. We have added this information to the figure legend.

Fig. S1 Why are the western blots from Fig. 1C shown again?

We are sorry for the redundancy. We have removed the western blots from Figure S1.

Depletion of the different cohesin subunits and regulator leads to a change in the cell cycle

distribution. For example, PDS5A and PDS5B depletion seem to strongly enrich cells in G1-phase. How does this impact the data generated with the different approaches as well as the data analysis?

We thank the reviewer for this important point. We agree that our samples have a small cell cycle variation due to asynchronous cells. However, we have verified that most of the cells are in G1 phase and that the cell cycle turns even after siRNA depletions in all samples. Also, our results of sample comparison using Hi-C and RNA-seq (Figures 2c, 4a, 5d, and 7e in the revised manuscript) did not show the clear correlation with cell cycle shown in Figure S1a. Therefore, we conclude that the subtle difference in cell cycle distribution among samples does not have that much impact. We hope the reviewer will agree with this perspective.

Fig. 7A The relevance of this figure is unclear. The figure legend is not informative.

We apologize for the inadequate explanation. Figure 7A in the original manuscript (Figure 8a in the revised manuscript) shows the peak-level comparison of depletion effects on long-range interactions by SIMA analysis. The network shows the depletion effect on chromatin interaction (edges) between all pairwise features, including ChIP-seq peaks and the TSSs of DEGs (nodes) for six depletions as representatives. Edge color and width indicate effect size and significance by Wilcoxon signed-rank test. We added this explanation in the figure legend.

line 110 – the authors evaluate similarity of the Hi-C samples. What are the criteria for this analysis?

The similarity of Hi-C samples (shown in Figure 1D in the original manuscript) is based on a stratum-adjusted correlation coefficient calculated by HiCRep (Yang *et al.*, *Genome Research* 2017). We adopted this score because it reflects the similarity of the TAD and loop structure involved in Hi-C data and shows high reproducibility among replicates.

Line 117 – the figure reference is not correct

We apologize for the lack of explanation. In Figure S2a, we show the averaged chromatin loop strength after depletion using the APA plot. By comparing siNIPBL samples for three different time points, we can see the insufficient depletion effect in the 24 h treatment. We have added the explanation in the figure legend of Figure S2a.

Line 131-132 – unclear whether the claim is really that TAD formation is strengthened by the depletion of cohesin and loaders

We apologize for the misleading explanation. What is strengthened by the depletion of cohesin and loaders is compartmentalization. In the revised manuscript, we have changed the sentence as follows.

“Such compartmentalization can be uncoupled from TAD formation; the compartmentalization is strengthened by the depletion of cohesin and loaders but not by depletion of CTCF.”

Line 366 – The authors refer to other studies using only single Hi-C samples. It is unclear what the authors intend to say - a single replicate or depletion of only one factor?

We apologize for the inadequate explanation. The other studies attempted to cluster genomic regions using a single Hi-C sample without replication, i.e., not by comparing samples. In contrast, our study clustered genomic regions based on the variation in depletion effects among samples. Our methodology is essential to capture the functional variation of cohesin and related factors in a context-specific manner. We have added this explanation in the revised manuscript (lines 388-392).

Reviewer #2:

Overview:

The article presents a comparative analysis of the effects of the knockdown of cohesin or related factors on chromatin folding, transcriptome, and histone modifications in human RPE cells. The authors employed a combination of in situ Hi-C, spike-in ChIP-seq, and RNA-seq data to investigate the impact of siRNA-depleted cellular context on these molecular mechanisms. The study initially examined the separate effects of factor depletion on TADs and loops, compartments, and DEGs using Hi-C, RNA-seq, and ChIP-seq data. Subsequently, the authors integrated these different patterns to analyze the effects on chromatin folding and transcriptome in a context-dependent manner. Notably, the author found a substantial depletion of interTAD interactions between “active TAD” and “inactive TAD”, which was region-specific and asymmetrical. These findings align with previous research while also presenting novel insights.

The authors’ presentation of their findings through the use of figures and their comprehensive analysis are noteworthy. However, there remain several areas that would benefit from further elaboration and clarification.

We are grateful to the reviewer for the consideration of our manuscript, and detailed feedback on our work.

Major:

“The comparative Hi-C analysis”:

1. it is observed from Figures 2B and 2C that the co-depletion of PDS5A and PDS5B results in a mild decrease in compartment strength (~10 Mbp). The description of the result and its possible reason could be further elaborated upon to enhance the novelty of the work.

We thank the reviewer for pointing this out. Because the PDS5 proteins function as a cohesin unloader, it is reasonable that inhibition of them weakened the compartmentalization, because loss of cohesin strengthens the compartmentalization (Rao *et al.*, *Cell* 2017 and Schwarzer *et al.*, *Nature* 2017). Since depletion of PDS5A or PDS5B alone did not show the similar effect, they may have partly redundant functions, as suggested in Wutz *et al.* (Wutz *et al.* *EMBO J* 2017). We have added this explanation (lines 148-152).

2. Page 7, line 141, the statement “After siRad21, most short loops were depleted, and the distribution peaked at a longer length (~10 Mbp) than the control (~500 kbp) ...” appears to be inconsistent with the data presented in Figure 2D. In this figure, the control peaked at approximately 400 kbp, while siRad21 only exhibited a slightly higher frequency around 800 kbp loops after losing short loops. Additionally, siCTCF displayed a peak at 500 kbp and the NIPBL-Rad21 co-depletion was less dramatic than siRad21. Therefore, it is recommended that the entire paragraph be reviewed and corrected.

We appreciate your clarification. The explanation in the original manuscript was incorrect. We have changed the paragraph according to the figure (lines 155-157). We have also added the line graph of the loop distribution corresponding to Figure 3d in Figure S4a to show the distribution more clearly.

“The comparative RNA-seq analysis”:

1. The correlation map presented in Figure 3A employs the Simpson index, which solely considers the number of overlapping DEGs across siRNA samples and disregards their expression levels. This raises the concern of introducing bias in the correlation map, as it treats up- and down-regulated DEGs equally. Therefore, it would be beneficial to address this issue and consider alternative methods that account for differential expression levels.

We thank the reviewer for this important point. Indeed, we also tried to classify RNA-seq samples based on the expression level of DEGs (Support Figure 1). The result was similar to the simple overlap analysis using the Simpson index for the top1000 DEGs (Figure 4a in the revised manuscript), suggesting that our clustering result is reasonable. We adopted the binary comparison in the RNA-seq analysis because we prepared the RNA-seq samples by three different experiments, each with control samples, the quantitative analysis would be more affected by the technical variation due to sample preparation. We have added this explanation in the Methods section (lines 610-614). Meanwhile, we have considered up- and down-regulated DEGs separately in the DEG clustering (Figure 4b in the revised manuscript), as suggested by the reviewer.

Support Figure 1: Correlation heatmap of RNA-seq samples based on z-score of the logged expression value.

2. Regarding Figure 3B, the description of the K-means clustering of DEGs is rather vague, and it would be helpful to clarify the input data and distance calculation method employed.

We apologize for the lack of explanation. For DEGs clustering, we generated DEG vectors of each siRNA that contain {1: upregulated, -1: downregulated, 0: not included} and created a matrix by combining the vectors of all depletions. We applied K-means clustering to the matrix and classified the DEGs into 20 clusters. We have added the description to the Methods section (lines 614-618).

3. In Figure 3C, the observed increase in the enrichment of Rad21 peaks in the siRad21 sample, particularly in clusters 0, 2, 5, 7, 10, 14, and 18, raises questions regarding the underlying cause. Therefore, it would be beneficial to discuss potential explanations for this phenomenon.

We thank the reviewer for the careful reading. Because the depletion efficiency by siRNA is not perfect, the stronger Rad21 peaks in control sample remain more after siRad21 (see Figure 4d in the revised manuscript for an example). The permutation test (Figure 4c in the revised manuscript) estimated the relative overlap frequency of the ChIP-seq peaks at the DEG loci for each cluster against the background (in this case, all DEG loci). Therefore, the clusters where Rad21 peaks are enriched under siRad21 indicate that the clusters containing more remained (i.e. stronger) Rad21 peaks at TSSs.

In clusters 0, 2, 5, 7, 10, 14, and 18, all clusters except cluster 2 can be explained as overlapping with clusters where acetylated cohesin (Smc3ac) is enriched, since cohesin binding is stabilized by acetylation (see also our response to the next comment). As cluster 2 is also enriched in Mau2 peaks, it would be a cohesin loading point and then there are strong cohesin sites, although not statistically significant. We added these explanations in the revised manuscript (lines 211-215).

4. On Page 9, line 195, the statement “Acetylated cohesin ... than non-acetylated sites (Figure 3C)” lacks clarity and would benefit from a more detailed explanation and an accompanying illustration.

We apologize for the insufficient explanation about the acetylated cohesin. A certain amount of cohesin on the genome is acetylated by ESCO1, which protects cohesin from release by WAPL, resulting in more stable binding of cohesin to the genome (Wutz, *et al.*, *Elife* 2020). Therefore, the

acetylated cohesin sites (Smc3ac) would be more persistent even under siNIPBL and siRad21 than non-acetylated sites. We have added this explanation in the revised manuscript (lines 208-211).

5. Figure 3D displays the top-ranked DEGs, and it would be informative to know whether they are up- or down-regulated after depletion and the potential reasons for such changes.

We appreciate your suggestion. As suggested by the reviewer, we have added the depletion in which the gene was up- or downregulated to Figure 4d in the revised manuscript. Regarding the potential reasons, we would like to make sure that the readers can speculate on the reason from the results themselves, as there are still multiple possibilities. We hope that the reviewer will agree with this perspective.

6. Figure 3E illustrates that siCTCF significantly increases cohesin peaks at upstream and exon. Please provide possible explanations.

This result is reminiscent of the report by Busslinger *et al.* that in CTCF knockout mouse embryonic fibroblast (MEF) cells cohesin accumulated near the TSSs of active genes, where the cohesin loader is also located (Busslinger *et al.*, *Nature* 2017). In the absence of CTCF, cohesin cannot properly accumulate at CTCF sites and would be more correlated with gene activity and cohesin loading sites. We have added this explanation in the revised manuscript (lines 225-229).

“The multi-scale insulation of TAD boundaries”:

1. In Figure 4D, the insulation score of TAD boundaries increased after depletion but was labeled as “Boundary loss (TAD fusion)”. I wonder if it’s wrong as the boundary appears to be strengthened.

We apologize for the lack of explanation. For the insulation score (y-axis), the larger the value, the less insulated (i.e. weakened boundaries). For the clarity, we have added an explanation for the y-axis in the figure in the revised manuscript.

2. On page 10, line 231, “resulting in the dysregulation of the expression of genes in the surrounding region”. However, the current evidence does not establish a clear relationship between the cohesin-separated boundaries and the position of the DEGs.

We appreciate for the important comment. We agree with that this is our speculation and have

modified the sentence as follows (lines 252-254): “This result suggests that the loss of cohesin loading at these points leads to an enhancement of insulation, which could potentially lead to dysregulation of gene expression in the surrounding region.”

3. On page 10, line 233, “would be more highly correlated with DEGs than ...”. It may be more appropriate to state that the insulation score is more highly correlated with “up-regulated DEGs”.

Thank you for the comment. We have corrected the sentence as you guided us.

“The cohesin distribution over AB compartment”:

1. On page 12, line 263, the author suggests that “there are different amounts of cohesin between compartment A and B”. which could be quantified and compared using Rad21 ChIP-seq of the control. I wonder if the amount difference of depleted cohesin between compartment A and B is due to the depletion bias in the siRNA system. If so, the statement could not reflect the amounts of cohesin in the control and should be changed to “different amount of affected cohesin”. As the same on page 12, line 269 “the cohesin accumulated to the highest levels at ...”, it might need to change to “the depleted cohesin accumulated to ...”.

Thank you for the insightful comment. The siRNA system targets mRNA and stops synthesizing new proteins, independent of genomic context. Therefore, there should be no bias in the siRNA effect between compartments A and B. Furthermore, siNIPBL also showed a comparable pattern to siRad21 (Figure 6a), suggesting that the decrease of cohesin on compartment A is not a technical bias of the target sequence of siRNA. Consequently, we think the decrease observed in Figure 6a is unbiased and represents the distribution of cohesin abundance in control samples. We have added this discussion in the Discussion section (lines 420-422).

Furthermore, I wonder if the biased affection on the A/B compartment is observed from the whole genome or just the illustrated part in Figure 5A, chr21: 24-32 Mbp?

We investigated the trend across the whole genome (shown in Figures 5B and 5D in the original manuscript, Figures 6b and 6d in the revised manuscript). We observed a more significant amount of cohesin in “strong A” regions in a genome-wide manner (Figure 6b in the revised manuscript). Our extended chromatin state analysis also showed that cohesin accumulated to the highest levels at highly active sites (states 4 and 7, enriched for H3K27ac and H3K4me3; Figure 6d in the revised manuscript). siCTCF did not show such a context-specific tendency. From these observations, we concluded that this trend is genome-wide. To further inform readers, we have

uploaded the genome-wide distribution of Figure 6a to Zenodo (Figure 6/pdf/DOWN.Penrich.8MB.5000_chr*.pdf).

2. On page 12 Line 266, the author states that “a milder loss of intraTAD interactions in Strong B than in Strong A regions, whereas siCTCF affected both.” However, it appears that there was a more severe loss of intraTAD interactions in the Strong B region in Figure 5C. The author should clarify and correct this consistency.

We apologize for the inadequate explanation. We meant to emphasize the difference in the depletion effect between siRad21 and siCTCF. Based on the reviewer's comment, we have changed the sentence to “This observation is also supported by the milder loss in Strong B and a severer loss in Strong A in intraTAD interactions under siRad21 compared to siCTCF.” (lines 288-290)

“Methods”:

On page 2, line 23, page 4, line 77, and page 16, line 360, the author mentioned the main component of their comparative method, CustardPy. However, the Methods section lacks a description of the function or models employed by CustardPy. An overview of CustardPy is deemed necessary.

Thank you for the important comment. In the revised manuscript, we have outlined what was done by CustardPy in the workflow (Figure 1a) and the Discussion section (lines 384-392). In the Methods section, we have added a new subsection “Hi-C analysis by CustardPy”. As shown in Figure 1a, almost all of the Hi-C analysis described in the Methods section is included in CustardPy.

Minor:

1. Please explain the statement on Page 3, line 57 “... suggesting that CTCF boundaries are not absolute.”

The mechanism of the loop extension under siWAPL has been suggested to be due to the "passing through" of CTCF sites due to temporal dissociation of CTCF (Haarhuis *et al.*, *Cell* 2017) or the clustering of neighboring CTCF boundaries called "traffic jam" (Allahyar *et al.*, *Nat Genet* 2018). Since such extended loops are rare but also occur in wild-type cells, the dynamics of TAD boundaries and loop formation is rather stochastic and depends on the amount of cohesin on

chromatin, which we called “CTCF boundaries are not absolute”. Since the sentence in the original manuscript was not sufficient, we have changed the sentences in the revised manuscript (lines 58-63).

2. On page 3, line 60, “Cohesin turnover is also critical for proper gene expression for regulation and for the CdLS phenotype”. Adding some examples will be much helpful.

We have added two examples in the revised manuscript (lines 63-67).

3. Page 3, line 67, “Moreover, cohesin and CTCF also localize within TADs without forming boundaries.” Please provide its reference.

We have added the references (Huang *et al.*, *Nature Genetics* 2021) and (Faure *et al.*, *Genome Research* 2012).

4. Page 9, line 203, “other regions” can be replaced by “BMP6’s exon, SSBP2’s intron, and the intergenic region”.

We have replaced it as the reviewer suggested.

5. The image resolution of Figure 5B (left) is not sufficient to show the thresholds clearly that separate the strong A region and weak A region.

We have improved the resolution of the figure.

Reviewer #3:

The manuscript "A comparative multi-omics analysis of context-dependent perturbations in chromatin folding and the transcriptome by cohesin and related factors" by Nakato et al. introduces a large multi-omics dataset of cohesin perturbation and a computational workflow, CustardPy, for multi-omics (replicated Hi-C, RNA-seq, spike-in ChIP-seq) data analysis (Python3 and two Docker images implementation, with documentation). The introduction briefly introduces the role of cohesin and the associated factors (loaders, unloaders, CTCF, and others). The method is applied to define similarities and differences in the effect of depletion of individual factors by siRNAs in human retinal pigment cells. Besides validating previous observations, the

authors report the association of gene expression changes with TAD splitting, imbalanced enrichment of cohesin binding in A and B compartments (A1-5, B1, B2), the differential effect of cohesin and loaders depletion on short- and long-range interactions within TADs, differential effects of CTCF and cohesin. Investigation of the allele-specific depletion effect on chromosome X. The authors use creative methods for data analysis, e.g., considering "Weak" and "strong" A/B compartments, directional relative frequency, classification of insulation boundaries into six subtypes. There are also novel methods for visualizing the multi-omics results, e.g., Figure S5B showing the proximity of DEGs to disrupted TAD boundaries. All datasets are well-documented, and all main and supplementary figures and tables are well-described. Methods used for data analyses are considered gold-standard (e.g., Juicer, Cooler for Hi-C, DESeq2 for RNA-seq). The CustardPy method is implemented in XYZ and wrapped in a Docker image.

We appreciate for your consideration of our manuscript, and detailed feedback on our work.

Minor

- The data is a big part of the submission. Suggest renaming files on GEO for better compliance with the FAIR principles. Currently, the file names are very heterogeneous, and hard to understand which one belongs to which technology/condition. Systematic file naming would facilitate finding the right data.

Thank you for the important suggestion. We have modified the name of our data files on the GEO database to be clearer.

- The Custardpy documentation describes both Docker and Singularity installations, However, the following usage examples are for Singularity only. It may be helpful to provide Docker syntax as the primary examples, supplemented with Singularity examples (as tabs or expandable sections).

Thank you for your suggestion. We have updated the documentation and added the NOTE first to support both Docker and Singularity.

- "Eigenvector (PC1) values for compartment analysis were calculated by HiC1Dmetrics (because the Eigenvector command in Juicer tools." - seems unfinished, and the HiC1Dmetrics is unreferenced.

Thank you for pointing this out. We have corrected the sentence and added the reference of

HiC1Dmetrics in the revised manuscript.

- Saddle plots are often difficult to quantify. Suggesting to use an approach used in Du *et al.*, “DNA Methylation Is Required to Maintain Both DNA Replication Timing Precision and 3D Genome Organization Integrity.”, Figure 4B-F (they provide code).

Thank you for the useful information. We have carefully reviewed the paper by Du *et al.* and found that they used GENOVA (<https://github.com/robinweide/GENOVA>) to calculate compartment strength from saddle plots. As suggested by the reviewer, in the revised manuscript we used GENOVA to calculate the compartment strength instead of our custom script and updated the result (Figure 3b in the revised manuscript). The relative distribution among the samples is almost the same as in the original manuscript.

- Stripe-like structures were detected. Tools started to appear to analyze them (FIREcaller, CHESS). It may be a good idea to use them for the sake of standardizing analyses. And, include it in the Docker images.

Thank you for your suggestion. We have carefully checked the suggested tools and found that FIREcaller (<https://github.com/yycunc/FIREcaller>) is designed to detect Frequently Interacting Regions (FIREs), not architectural stripes. CHESS (<https://chess-hic.readthedocs.io/>) is a great tool for the quantitative comparison and automatic feature extraction using a signal-to-noise metric. Although the original manuscript of CHESS describes that it can be used to analyze stripe structure, unfortunately there is no explanation of how to analyze it in the CHESS documentation. Meanwhile, we agree that FIREcaller and CHESS would add more insight to our results due to their sophisticated approach, and we would like to address this in the near future. To this end, we have added FIREcaller and CHESS to the latest version of the CustardPy docker image (version 1.2.0), as suggested by the reviewer.

REVIEWERS' COMMENTS

Reviewer #1 (Remarks to the Author):

The authors have revised the manuscript thoroughly and addressed all the points raised. The manuscript significantly improved and would be suitable for publication. Just a small remark, the coloring of the dots in Figure 3B is still confusing. The protein names on the y-axis have one color but the corresponding data another one without a reference for the meaning of the color. The note added in the legend is not sufficient.

Reviewer #2 (Remarks to the Author):

All my comments/questions have been addressed.

Reviewer #3 (Remarks to the Author):

All comments have been addressed.

Reviewer #1 (Remarks to the Author):

The authors have revised the manuscript thoroughly and addressed all the points raised. The manuscript significantly improved and would be suitable for publication.

Just a small remark, the coloring of the dots in Figure 3B is still confusing. The protein names on the y-axis have one color but the corresponding data another one without a reference for the meaning of the color. The note added in the legend is not sufficient.

We are grateful to the reviewer for the consideration of our manuscript, and detailed feedback on our work. We have revised the Figure 3b to make the colors consistent with the labels. We hope this is satisfactory.

Reviewer #2 (Remarks to the Author):

All my comments/questions have been addressed.

We are grateful to the reviewer for the consideration of our manuscript, and detailed feedback on our work.

Reviewer #3 (Remarks to the Author):

All comments have been addressed.

We are grateful to the reviewer for the consideration of our manuscript, and detailed feedback on our work.